# More ConvNets in the 2020s: Scaling up Kernels Beyond 51 × 51 using Sparsity

**Shiwei Liu[1,2], Tianlong Chen[1]\*Xiaohan Chen[1]\*Xuxi Chen[1], Qiao Xiao[2], Boqian Wu[2],**
**Tommi Kärkkäinen[4], Mykola Pechenizkiy[2], Decebal Constantin Mocanu[2,3,5], Zhangyang Wang[1]**
[1]University of Texas at Austin, [2]Eindhoven University of Technology,
[3]University of Twente, [4]University of Jyväskylä, [5]University of Luxembourg

Codes: https://github.com/VITA-Group/SLaK

## Abstract

Transformers have quickly shined in the computer vision world since the emergence of Vision Transformers (ViTs). The dominant role of convolutional neural networks (CNNs) seems to be challenged by increasingly effective transformer-based models. Very recently, a couple of advanced convolutional models strike back with large kernels motivated by the local-window attention mechanism, showing appealing performance and efficiency. While one of them, i.e. RepLKNet, impressively manages to scale the kernel size to $31 \times 31$ with improved performance, the performance starts to saturate as the kernel size continues growing, compared to the scaling trend of advanced ViTs such as Swin Transformer. In this paper, we explore the possibility of training extreme convolutions larger than $31 \times 31$ and test whether the performance gap can be eliminated by strategically enlarging convolutions. This study ends up with a recipe for applying extremely large kernels from the perspective of sparsity, which can smoothly scale up kernels to $61 \times 61$ with better performance. Built on this recipe, we propose *Sparse Large Kernel Network* (**SLaK**), a pure CNN architecture equipped with sparse factorized $51 \times 51$ kernels that can perform on par with or better than state-of-the-art hierarchical Transformers and modern ConvNet architectures like ConvNeXt and RepLKNet, on ImageNet classification as well as a wide range of downstream tasks including semantic segmentation on ADE20K, object detection on PASCAL VOC 2007, and object detection/segmentation on MS COCO.

## 1 Introduction

Since invented (Fukushima & Miyake, 1982; LeCun et al., 1989; 1998), convolutional neural networks (CNNs) (Krizhevsky et al., 2012a; Simonyan & Zisserman, 2015.; He et al., 2016; Huang et al., 2017; Howard et al., 2017; Xie et al., 2017; Tan & Le, 2019) have quickly evolved as one of the most indispensable architectures of machine learning in the last decades. However, the dominance of CNNs has been significantly challenged by Transformer (Vaswani et al., 2017) over the past few years. Stemming from natural language processing, Vision Transformers (ViTs) (Dosovitskiy et al., 2021; d'Ascoli et al., 2021; Touvron et al., 2021b; Wang et al., 2021b; Liu et al., 2021e; Vaswani et al., 2021) have demonstrated strong results in various computer vision tasks including image classification (Dosovitskiy et al., 2021; Yuan et al., 2021b), object detection (Dai et al., 2021; Liu et al., 2021e), and segmentation (Xie et al., 2021; Wang et al., 2021a;c; Cheng et al., 2021). Meanwhile, works on understanding of ViTs have blossomed. Plausible reasons behind the success of ViTs are fewer inductive bias (Dosovitskiy et al., 2021), long-range dependence (Vaswani et al., 2017), advanced architecture (Yu et al., 2021), and more human-like representations (Tuli et al., 2021), etc.

Recently, there is a rising trend that attributes the supreme performance of ViTs to the ability to capture a large receptive field. In contrast to CNNs which perform convolution in a small sliding window (e.g., $3 \times 3$ and $5 \times 5$) with shared weights, global attention or local attention with larger window sizes in ViTs (Liu et al., 2021e) directly enables each layer to capture large receptive field. Inspired by this trend, some recent works on CNNs (Liu et al., 2022b; Ding et al., 2022) strike back by designing advanced pure CNN architecture and plugging large kernels into them. For instance, RepLKNet (Ding et al., 2022) successfully scales the kernel size to $31 \times 31$, while achieving

---

\*Equal contribution.

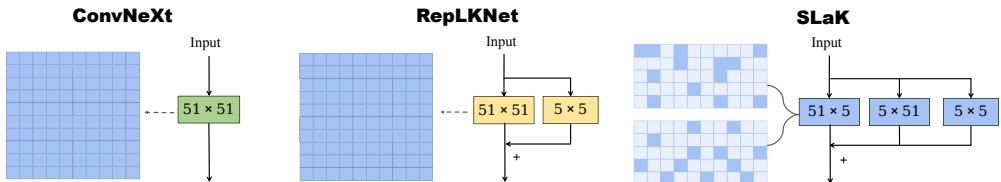

Figure 1: Large depth-wise kernel (e.g., 51×51) paradigms of ConvNeXt, RepLKNet, and SLaK. Dark blue squares refer to the dense weights in convolutional kernels. Light blue squares refer to the sparse weights in convolutional kernels.

comparable results to Swin Transformer (Liu et al., 2021e). However, large kernels are notoriously difficult to train. Even with the assistance of a parallel branch with small kernels, the performance of RepLKNet starts to saturate as the kernel size continues increasing, compared to the scaling trend of advanced ViTs such as Swin Transformer. Therefore, it remains mysterious whether we can exceed the Transformer-based models by further scaling the kernel size beyond 31×31.

In this paper, we attempt to answer this research question by leveraging *sparsity* commonly observed in the human visual system. Sparsity has been seen as one of the most important principles in the primary visual cortex (V1) (Tong, 2003), where the incoming stimuli have been hypothesized to be sparsely coded and selected (Desimone & Duncan, 1995; Olshausen & Field, 1997; Vinje & Gallant, 2000). We extensively study the trainability of large kernels and unveil three main observations: **(i)** existing methods that either naively apply larger kernels (Liu et al., 2022b) or assist with structural re-parameterization (Ding et al., 2022) fail to scale kernel sizes beyond 31×31; **(ii)** replacing one large M×M kernel with two rectangular, parallel kernels (M×N and N×M, where N < M) can smoothly scale the kernel size up to 61×61 with improved performance; **(iii)** constructing with sparse groups while expanding width significantly boosts the performance.

Built upon these observations, we propose SLaK – Sparse Large Kernel Network – a new pure CNN architecture equipped with an unprecedented kernel size of 51×51. Evaluated across a variety of tasks including ImageNet classification (Deng et al., 2009), semantic segmentation on ADE20K (Zhou et al., 2019), object detection on PASCAL VOC 2007 (Everingham et al., 2007), and object detection/segmentation on COCO (Lin et al., 2014), SLaK performs better than or on par with CNN pioneers RepLKNet and ConvNeXt (Liu et al., 2022b) as well as SOTA attention-based models e.g., Swin (Liu et al., 2021e) and Cswin (Dong et al., 2022) Transformers on ImageNet. Our analysis of effective receptive field (ERF) shows that when plugged in the recently proposed ConvNeXt, our method is able to cover a large ERF region than existing larger kernel paradigms.

## 2  RELATED WORK

**Large Kernel in Attention.** Originally introduced for Natural Language Processing (Vaswani et al., 2017) and extended in Computer Vision by Dosovitskiy et al. (2021), self-attention can be viewed as a global depth-wise kernel that enables each layer to have a global receptive field. Swin Transformer (Liu et al., 2021e) is a ViTs variant that adopts local attention with a shifted window manner. Compared with global attention, local attention (Ramachandran et al., 2019; Vaswani et al., 2021; Chu et al., 2021; Liu et al., 2021d; Dong et al., 2022) can greatly improve the memory and computation efficiency with appealing performance. Since the size of attention windows is at least 7, it can be seen as an alternative class of large kernel. A recent work (Guo et al., 2022b) proposes a novel large kernel attention module that uses stacked depthwise, small convolution, dilated convolution as well as pointwise convolution to capture both local and global structure.

**Large Kernel in Convolution.** Large kernels in convolution date back to the 2010s (Krizhevsky et al., 2012b; Szegedy et al., 2015; 2017), if not earlier, where large kernel sizes such as 7×7 and 11×11 are applied. Global Convolutional Network (GCNs) (Peng et al., 2017) enlarges the kernel size to 15 by employing a combination of 1×M + M×1 and M×1 + 1×M convolutions. However, the proposed method leads to performance degradation on ImageNet. The family of Inceptions (Szegedy et al., 2016; 2017) allows for the utilization of varying convolutional kernel sizes to learn spatial patterns at different scales. With the popularity of VGG (Simonyan & Zisserman, 2014), it has been common over the past decade to use a stack of small kernels (1×1 or 3×3) to obtain a large receptive field (He et al., 2016; Howard et al., 2017; Xie et al., 2017; Huang et al., 2017). Until very recently, some

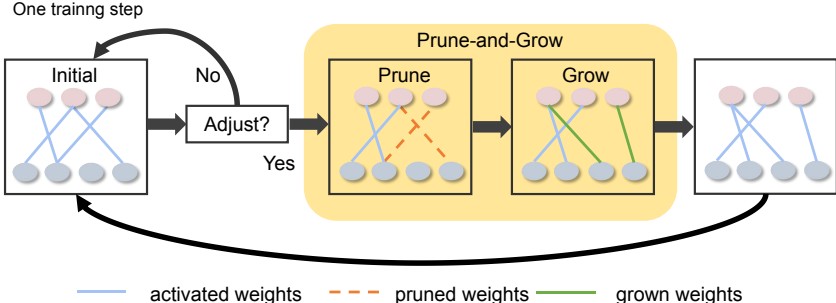

Figure 2: **Dynamic sparsity.** Dynamic sparsity allows us to construct and train initially sparse neural networks (sparse kernels) from scratch. During training, it dynamically adjusts the sparse weights by pruning the least important weights and adding new. Such dynamic procedure gradually optimizes the sparse kernels to a good pattern and hence encourages a more elaborate capture of local features.

works start to revive the usage of large kernels in CNNs. Li et al. (2021) propose involution with 7×7 large kernels that uses distinct weights in the spatial extent while sharing weights across channels. However, the performance improvement plateaus when further expanding the kernel size. Han et al. (2021b) find that dynamic depth-wise convolution (7×7) performs on par with the local attention mechanism if we substitute the latter with the former in Swin Transformer. Liu et al. (2022b) imitate the design elements of Swin Transformer (Liu et al., 2021e) and design ConvNeXt employed with 7×7 kernels, surpassing the performance of the former. RepLKNet (Ding et al., 2022) for the first time scales the kernel size to 31×31 by constructing a small kernel (e.g., 3×3 or 5×5) parallel to it and achieves comparable performance to the Swin Transformer. A series of work (Romero et al., 2021; 2022) of continuous convolutional kernels can be used on data of arbitrary resolutions, lengths and dimensionalities. Lately, Chen et al. (2022) reveal large kernels to be feasible and beneficial for 3D networks too. Prior works have explored the idea of paralleling (Peng et al., 2017; Guo et al., 2022a) or stacking (Szegedy et al., 2017) two complementary M×1 and 1×M kernels. However, they limit the shorter edge to 1 and do not scale the kernel size beyond 51×51. Different from those prior arts, we decompose a large kernel into two complementary non-square kernels (M×N and N×M), improving the training stability and memory scalability of large convolutions kernels.

**Dynamic Sparsity.** A long-standing research topic, recent attempts on sparsity (Mocanu et al., 2018; Liu et al., 2021b;c; Evci et al., 2020; Mostafa & Wang, 2019; Dettmers & Zettlemoyer, 2019; Chen et al., 2021) train intrinsically sparse neural networks from scratch using only a small proportion of parameters and FLOPs (as illustrated in Figure 2). Dynamic sparsity enables training sparse models from scratch, hence the training and inference FLOPs and memory requirements are only a small fraction of the dense models. Different from post-training pruning (Han et al., 2015; Frankle & Carbin, 2019), models built with dynamic sparsity can be trained from scratch to match their dense counterparts without involving any pre-training or dense training. Dynamic sparsity stems from Sparse Evolutionary Training (SET) (Mocanu et al., 2018; Liu et al., 2021b) which randomly initializes the sparse connectivity between layers randomly and dynamically adjusts the sparse connectivity via a parameter prune-and-grow scheme during the course of training. The parameter prune-and-grow scheme allows the model's sparse structure to gradually evolve, achieving better performance than naively training a static sparse network (Liu et al., 2021c). In this paper, our target is not to find sparse networks that can match the corresponding dense networks. Motivated by the principle of ResNeXt (Xie et al., 2017; Liu et al., 2022b) – "use more groups, expand width", we instead attempt to leverage dynamic sparsity to scale neural architectures with extreme kernels.

## 3    FAILURES OF EXISTING APPROACHES TO GO BEYOND 31×31 KERNELS

We first study the performance of extreme kernel sizes larger than 31×31 using two existing large-kernel techniques, ConvNeXt (Liu et al., 2022b) and RepLKNet (Ding et al., 2022). We take the recently-developed ConvNeXt on ImageNet-1K as our benchmark to conduct this study. We adopt the efficient large-kernel implementation developed by MegEngine (Meg, 2020) in this paper.

We follow recent works (Liu et al., 2022b; Bao et al., 2021; Liu et al., 2021e; Ding et al., 2022; Touvron et al., 2021b) using Mixup (Zhang et al., 2017), Cutmix (Yun et al., 2019),

Table 1: Test accuracy of 120-epoch ConvNeXt-T trained with various large kernel recipes on ImageNet-1K. "Naive" refers to directly enlarging kernel size of ConvNeXt; "RepLKNet" refers to training ConvNeXt with structural re-parameterization (Ding et al., 2022). The original ConvNeXt is built with 7×7 kernels.

| Kernel Size | Top-1 Acc | #Params | FLOPs | Top-1 Acc | #Params | FLOPs |
|---|---|---|---|---|---|---|
| | Naive | | | RepLKNet | | |
| 7-7-7-7 | 81.0 | 29M | 4.5G | | | |
| 31-29-27-13 | 80.5 | 32M | 6.5G | 81.5 | 32M | 6.1G |
| 51-49-47-13 | 80.4 | 38M | 9.4G | 81.3 | 38M | 9.3G |
| 61-59-57-13 | 80.3 | 43M | 11.6G | 81.3 | 43M | 11.5G |

RandAugment (Cubuk et al., 2020), and Random Erasing (Zhong et al., 2020) as data augmentations. Stochastic Depth (Huang et al., 2016) and Label Smoothing (Szegedy et al., 2016) are applied as regularization with the same hyper-parameters as used in ConvNeXt. We train models with AdamW (Loshchilov & Hutter, 2019). **It is important to note** that all models are trained for *a reduced length of 120 epochs* in this section, just to sketch the scaling trends of large kernel sizes. Later in Section 5, we will adopt the full training recipe and train our models for 300 epochs, to enable fair comparisons with state-of-the-art models. Please refer to Appendix A for more details.

Liu et al. (2022b) show that naively increasing kernel size from 3×3 to 7×7 brings considerably performance gains. Very recently, RepLKNet (Ding et al., 2022) successfully scales convolutions up to 31×31 with structural re-parameterization (Ding et al., 2019; 2021). We further increase the kernel size to 51×51 and 61×61 and see whether larger kernels can bring more gains. Following the design in RepLKNet, we set the kernel size of each stage as [51, 49, 47, 13] and [61, 59, 57, 13], and report test accuracies in Table 1. As expected, naively enlarging kernel size from 7×7 to 31×31 decreases the performance, whereas RepLKNet can overcome this problem by improving the accuracy by 0.5%. Unfortunately, this positive trend does not continue when we further increase kernel size to 51×51.

One plausible explanation is that although the receptive field may be enlarged by using extremely large kernels, it might fail to maintain the desirable property of locality. Since the stem cell in standard ResNet (He et al., 2016) and ConvNeXt results in a 4× downsampling of the input images, extreme kernels with 51×51 are already roughly equal to the global convolution for the typical 224 × 224 ImageNet. Therefore, this observation makes sense as well-designed local attention (Liu et al., 2021e;d; Chu et al., 2021) usually outperforms global attention (Dosovitskiy et al., 2021) in a similar mechanism of ViTs. Motivated by this, we see the opportunity to address this problem by introducing the locality while preserving the ability to capture global relations.

## 4   A RECIPE FOR EXTREMELY LARGE KERNELS BEYOND 31×31

In this section, we introduce a simple, two-step recipe for extremely large kernels beyond 31×31:

> **Step 1.** Decomposing a large kernel into two rectangular, parallel kernels.
> **Step 2.** Using *sparse* groups, expanding more width.

**Decomposing a large kernel into two rectangular, parallel kernels smoothly scales the kernel size up to 61×61.** Although using convolutions with medium sizes (e.g., 31×31) seemingly can directly avoid this problem, we want to investigate if we can further push the performance of CNNs by using (global) extreme convolutions. Our recipe here is to approximate the large M×M kernel with a combination of two parallel and rectangular convolutions whose kernel size is M×N and N×M (where N < M), respectively, as shown in Figure 1. Following Ding et al. (2022), we keep a 5×5 layer parallel to the large kernels and summed up their outputs after a batch norm layer.

This decomposition balances between capturing long-range dependencies and extracting local detail features (with its shorter edge). Moreover, existing techniques for large kernel training (Liu et al., 2022b; Ding et al., 2022) suffer from quadratic computational and memory overhead as the kernel size increases. In stark contrast, the overhead of our method increases just linearly with the kernel size (Figure 4). The performance of kernel decomposition with N = 5 (see Appendix E for the effect of N) is reported as the "Decomposed" group in Table 2. As the decomposition reduces learnable

parameters and FLOPs, it is no surprise to observe our network to initially sacrifice accuracy slightly compared to the original RepLKNet at medium kernel sizes i.e. $31\times31$. However, as the convolution size continues to increase, our method can *scale* kernel size up to $61\times61$ with improved performance.

Table 2: Test accuracy of ConvNeXt-T trained with various large kernel recipes on ImageNet-1K. All the models are trained for 120 epochs.

| Kernel Size | Top-1 Acc | #Params | FLOPs | Top-1 Acc | #Params | FLOPs | Top-1 Acc | #Params | FLOPs |
|---|---|---|---|---|---|---|---|---|---|
| | Decomposed | | | Sparse groups | | | Sparse groups, expand more width | | |
| 7-7-7-7 | 81.0 | 29M | 4.5G | 80.0 | 17M | 2.6G | 81.1 | 29M | 4.5G |
| 31-29-37-13 | 81.3 | 30M | 5.0G | 80.4 | 18M | 2.9G | 81.5 | 30M | 4.8G |
| 51-49-47-13 | 81.5 | 31M | 5.4G | 80.5 | 18M | 3.1G | 81.6 | 30M | 5.0G |
| 61-59-57-13 | 81.4 | 31M | 5.6G | 80.4 | 19M | 3.2G | 81.5 | 31M | 5.2G |

**"Use *sparse* groups, expand more width" significantly boosts the model capacity.** Recently proposed ConvNeXt (Liu et al., 2022b) revisits the principle introduced in ResNeXt (Xie et al., 2017) that splits convolutional filters into small but more groups. Instead of using the standard group convolution, ConvNeXt simply employs depthwise convolutions with an increased width to achieve the goal of "use more groups, expand width". In this paper, we attempt to extend this principle from a sparsity-inspired perspective – "use *sparse* groups, expand more width".

To be specific, we first replace the dense convolutions with sparse convolutions, where the sparse kernels are randomly constructed based on the layer-wise sparsity ratio of SNIP (Lee et al., 2019)[1] due to its strong performance on large-scale models (Liu et al., 2022a). After construction, we train the sparse model with dynamic sparsity (Mocanu et al., 2018; Liu et al., 2021b), where the sparse weights are dynamically adapted during training by pruning the weights with the lowest magnitude and growing the same number of weights randomly. Doing so enables dynamic adaptation of sparse weights, leading to better local features. As kernels are sparse throughout training, the corresponding parameter count and training/inference FLOPs are only proportional to the dense models. See Appendix B for the full details of dynamic sparsity. To evaluate, we sparsify the decomposed kernels with 40% sparsity and report the performance as the "Sparse groups" column. We can observe in the middle column of Table 2 that dynamic sparsity notably reduces more than 2.0 GFLOPs, despite causing temporary performance degradation.

We next show that the above high efficiency of dynamic sparsity can be effectively transferred to model scalability. Dynamic sparsity allows us to computation-friendly scale the model size up. For example, using the same sparsity (40%), we can expand the model width by $1.3\times$ while keeping the parameter count and FLOPs roughly the same as the dense model. This brings us significant performance gains, increasing the performance from 81.3% to 81.6% with extreme $51\times51$ kernels. Impressively, equipped with $61\times61$ kernels, our method outperforms the previous state of the arts (Liu et al., 2022b; Ding et al., 2022) while saving 55% FLOPs.

**Large Kernels Generalize Better than Small Kernels with Our Recipe.** To demonstrate that the benefits of large kernels, we also report the impact of each step for the small $7\times7$ kernel in Table 2. We can clearly see that the performance consistently increases with kernel size, up to $51\times51$. Applying each part of our proposed recipe to $7\times7$ kernels leads to either no gain or marginal gains compared to our 51x51 kernels. This break-down experiment justifies our claim: large kernel is the root of power, and our proposed recipe helps unleash such power from large kernels.

## 4.1 BUILDING THE SPARSE LARGE KERNEL NETWORK (SLaK)

So far, we have discovered our recipe which can successfully scale up kernel size to $51\times51$ without backfiring performance. Built on this recipe, we next construct our own Sparse Large Kernel Network (SLaK), a pure CNN architecture employed with extreme $51\times51$ kernels. SLaK is built based on the architecture of ConvNeXt. The design of the stage compute ratio and the stem cell are inherited from ConvNeXt. The number of blocks in each stage is [3, 3, 9, 3] for SLaK-T and [3, 3, 27, 3] for SLaK-S/B. The stem cell is simply a convolution layer with $4\times4$ kernels and 4 strides.

We first directly increase the kernel size of ConvNeXt to [51, 49, 47, 13] for each stage, and replace each M×M kernel with a combination of M×5 and 5×M kernels as illustrated in Figure 1. We find

---

[1] SNIP ratio is obtained by globally selecting the important weights across layers with the highest connection sensitivity score $|g \odot w|$, where $w$ and $g$ is the network weight and gradient, respectively.

that adding a BatchNorm layer directly after each decomposed kernel is crucial before summing the output up. Following the guideline of "use *sparse* groups, expand more width", we further sparsify the whole network and expand the width of stages by 1.3×, ending up with SLaK. Even there could be a large room to improve SLaK performance by tuning the trade-off between model width and sparsity (as shown in Appendix D), we keep one set of hyperparameters (1.3× width and 40% sparsity) for all experiments, so SLaK works simply "out of the box" with no ad-hoc tuning at all.

## 5  EVALUATION OF SLaK

To comprehensively verify the effectiveness of SLaK, we compare it with various state-of-the-art baselines on a large variety of tasks, including: ImageNet-1K classification (Deng et al., 2009), semantic segmentation on ADE20K (Zhou et al., 2019), object detection on PASCAL VOC 2007 (Everingham et al., 2007), and object detection/segmentation on COCO.

Table 3: **Classification accuracy on ImageNet-1K.** For SLaK models, we report both theoretical, sparsity-aware numbers parameter & FLOPs (in black color), as well as those numbers measured if assuming no sparsity-aware acceleration (in blue color).

| Model | Image Size | #Param. | FLOPs | Top-1 Acc |
|---|---|---|---|---|
| ResNet-50 (He et al., 2016) | 224×224 | 26M | 4.1G | 76.5 |
| ResNeXt-50-32×4d (Xie et al., 2017) | 224×224 | 25M | 4.3G | 77.6 |
| ResMLP-24 (Touvron et al., 2021a) | 224×224 | 30M | 6.0G | 79.4 |
| DeiT-S (Touvron et al., 2021b) | 224×224 | 22M | 4.6G | 79.8 |
| Swin-T (Liu et al., 2021e) | 224×224 | 28M | 4.5G | 81.3 |
| TNT-S (Han et al., 2021a) | 224×224 | 24M | 5.2G | 81.3 |
| T2T-ViT$_t$-14 (Yuan et al., 2021a) | 224×224 | 22M | 6.1G | 81.7 |
| ConvNeXt-T (Liu et al., 2022b) | 224×224 | 29M | 4.5G | 82.1 |
| **SLaK-T** | 224×224 | 30M/50M | 5.0G/8.7G | **82.5** |
| Mixer-B/16 (Tolstikhin et al., 2021) | 224×224 | 59M | 11.6G | 76.4 |
| ResNet-101 (He et al., 2016) | 224×224 | 45M | 7.9G | 77.4 |
| ResNeXt101-32x4d (Xie et al., 2017) | 224×224 | 44M | 8.0G | 78.8 |
| PVT-Large (Wang et al., 2021b) | 224×224 | 61M | 9.8G | 81.7 |
| T2T-ViT$_t$-19 (Yuan et al., 2021a) | 224×224 | 39M | 9.8G | 82.4 |
| Swin-S (Liu et al., 2021e) | 224×224 | 50M | 8.7G | 83.0 |
| ConvNeXt-S (Liu et al., 2022b) | 224×224 | 50M | 8.7G | 83.1 |
| **SLaK-S** | 224×224 | 55M/91M | 9.8G/16.7G | **83.8** |
| DeiT-Base/16 (Touvron et al., 2021b) | 224×224 | 87M | 17.6G | 81.8 |
| RepLKNet-31B (Ding et al., 2022) | 224×224 | 79M | 15.3G | 83.5 |
| Swin-B (Liu et al., 2021e) | 224×224 | 88M | 15.4G | 83.5 |
| ConvNeXt-B (Liu et al., 2022b) | 224×224 | 89M | 15.4G | 83.8 |
| **SLaK-B** | 224×224 | 95M/158M | 17.1G/28.5G | **84.0** |
| ViT-Base/16 (Dosovitskiy et al., 2021) | 384×384 | 87M | 55.4G | 77.9 |
| DeiT-B/16 (Touvron et al., 2021b) | 384×384 | 86M | 55.4G | 83.1 |
| Swin-B (Liu et al., 2021e) | 384×384 | 88M | 47.1G | 84.5 |
| RepLKNet-31B (Ding et al., 2022) | 384×384 | 79M | 45.1G | 84.8 |
| ConvNeXt-B (Liu et al., 2022b) | 384×384 | 89M | 45.0G | 85.1 |
| **SLaK-B** | 384×384 | 95M/158M | 50.3G/83.8G | **85.5** |

### 5.1  EVALUATION ON IMAGENET-1K

ImageNet-1K contains 1,281,167 training images, 50,000 validation images. We use exactly the same training configurations in Section 4, except now training for the full 300 epochs, following ConvNeXt and Swin Transformer. We observed that models with BatchNorm layers and EMA see poor performance when trained over 300 epochs (also pointed by Liu et al. (2022b)), and resolved this by running one additional pass over the training data, the same as used in Garipov et al. (2018); Izmailov et al. (2018). Please refer to Appendix A for more details about the training configurations.

We compare the performance of SLaK on ImageNet-1K with various the state-of-the-arts in Table 3. With similar model sizes and FLOPs, SLaK outperforms the existing convolutional models such as ResNe(X)t (He et al., 2016; Xie et al., 2017), RepLKNet (Ding et al., 2022), and ConvNeXt (Liu et al., 2022b). Without using any complex attention modules and patch embedding, SLaK is able to achieve higher accuracy than the state-of-the-art transformers, e.g., Swin Transformer (Liu et al.,

2021e) and Pyramid Vision Transformer (Wang et al., 2021b; 2022). Perhaps more interestingly, directly replacing the 7×7 of ConvNeXt-S to 51×51 is able to improve the accuracy over the latter by 0.7%. Moreover, Table 3 shows that our model benefits more from larger input sizes: the performance improvement of SLaK-B over ConvNeXt-B on 384×384 is twice that on 224×224 input, highlighting the advantages of using large kernels on high-resolution training (Liu et al., 2021d).

Moreover, we also examine if SLaK can rival stronger Transformer models – CSwin Transformer (Dong et al., 2022), a carefully designed hybrid architecture with transformers and convolutions. CSwin performs self-attention in both horizontal and vertical stripes. As shown in Appendix C, SLaK consistently performs on par with CSwin Transformers, and is the first pure ConvNet model that can achieve so without any bells and whistles, to our best knowledge.

## 5.2  EVALUATION ON ADE20K

For semantic segmentation, we choose the widely-used ADE20K, a large-scale dataset which contains 20K images of 150 categories for training and 2K images for validation. To draw a solid conclusion, we conduct experiments with both short and long training procedures. The backbones are pre-trained on ImageNet-1K with 224×224 input for 120/300 epochs and then are finetuned with UperNet (Xiao et al., 2018) for 80K/160K iterations, respectively. We report the mean Intersection over Union (mIoU) with single-scale testing in Table 4 for comparison. The upper part of Table 4 demonstrates a very clear trend that the performance increases as the kernel size. Specifically, RepLKNet scales the kernel size of ConvNeXt-T to 31×31 and achieves 0.4% higher mIoU. Notably, SLaK-T with larger kernels (51×51) further brings 1.2% mIoU improvement over ConvNeXt-T (RepLKNet), surpassing the performance of ConvNeXt-S. With longer training procedures, SLaK also consistently outperforms its small-kernel baseline (ConvNeXt) by a good margin, reaffirming the advantages of large kernels on downstream dense vision tasks.

Table 4: **Semantic segmentation on ADE20K.** Models are pre-trained on ImageNet-1K with 224×224 input and finetuned on with UperNet. Following Ding et al. (2022), we report mIoU results with single-scale testing. FLOPs are based on input sizes of (2048, 512). Methods marked with * are implemented/reproduced by us.

| Model | Kernel Size | mIoU (↑) | #Param | FLOPs |
|---|---|---|---|---|
| pre-trained for 120 epochs, finetuned for 80K iteration | | | | |
| ConvNeXt-T (Liu et al., 2022b) | 7-7-7-7 | 44.6 | 60M | 939G |
| ConvNeXt-S (Liu et al., 2022b) | 7-7-7-7 | 45.9 | 82M | 1027G |
| ConvNeXt-T (RepLKNet)* (Ding et al., 2022) | 31-29-27-13 | 45.0 | 64M | 973G |
| SLaK-T | 51-49-47-13 | **46.2** | 65M | 936G |
| pre-trained for 300 epochs, finetuned for 160K iteration | | | | |
| ConvNeXt-T (Liu et al., 2022b) | 7-7-7-7 | 46.0 | 60M | 939G |
| SLaK-T | 51-49-47-13 | **47.6** | 65M | 936G |
| ConvNeXt-S (Liu et al., 2022b) | 7-7-7-7 | 48.7 | 82M | 1027G |
| SLaK-S | 51-49-47-13 | **49.4** | 91M | 1028G |
| ConvNeXt-B (Liu et al., 2022b) | 7-7-7-7 | 49.1 | 122M | 1170G |
| SLaK-B | 51-49-47-13 | **50.2** | 135M | 1172G |

## 5.3  EVALUATION ON PASCAL VOC 2007

PASCAL VOC 2007 is another popular benchmark for object detection and semantic segmentation. In this section, we evaluate our model on object detection. Models are on pre-trained on ImageNet-1K for 300 epochs and finetuned with Faster-RCNN (Ren et al., 2015). Table 5 shows the results comparing SLaK-T, ConvNeXt-T, ConvNet-T (RepLKNet), and traditional convolutional networks, i.e., ResNet. Again, large kernels lead to better performance. Specifically, ConvNeXt-T with 31×31 kernels achieves 0.7% higher mean Average Precision (mAP) than the 7×7 kernels and SLaK-T with 51 kernel sizes further brings 1.4% mAP improvement, highlighting the crucial role of extremely large kernels on downstream vision tasks.

## 5.4  EVALUATION ON COCO

Furthermore, we finetune Cascade Mask R-CNN (Cai & Vasconcelos, 2018) on MS COCO (Lin et al., 2014) with our SLaK backbones. Following ConvNeXt, we use the multi-scale setting and train models with AdamW. Please refer to Appendix A.3 for more implementation details. Again, we

Table 5: **Object detection on PASCAL VOC 2007.** Faster RCNN is equipped with various backbone networks that are pre-trained for 120 epochs on ImageNet-1K. The pre-trained ConvNeXt-T is obtained from its GitHub repository. FLOPs are based on input sizes of (1280, 800). Methods marked with * are implemented by us.

| Model | Kernel Size | mAP (%) ($\uparrow$) | #Param | FLOPs |
|---|---|---|---|---|
| ResNet-50 (He et al., 2016) | 3-3-3-3 | 74.0 | - | - |
| ResNet-101 (He et al., 2016) | 3-3-3-3 | 74.3 | - | - |
| ConvNeXt-T (Liu et al., 2022b) | 7-7-7-7 | 80.6 | 45M | 208G |
| ConvNeXt-T (RepLKNet)* (Ding et al., 2022) | 31-29-27-13 | 81.3 | 55M | 207G |
| SLaK-T | 51-49-47-13 | **82.7** | 49M | 205G |

see an encouraging trend: the performance consistently improves with the increase of kernel size and our $51 \times 51$ kernel SLaK outperforms smaller-kernel models.

Table 6: **Object detection and segmentation on MS COCO.** Models are pre-trained on ImageNet-1K and finetuned using Cascade Mask-RCNN. Methods marked with * are implemented by us.

| Model | Kernel Size | $AP^{box}$ | $AP^{box}_{50}$ | $AP^{box}_{75}$ | $AP^{mask}$ | $AP^{mask}_{50}$ | $AP^{mask}_{75}$ |
|---|---|---|---|---|---|---|---|
| pre-trained for 120 epochs, finetuned for $1\times$ (12 epochs) | | | | | | | |
| ConvNeXt-T (Liu et al., 2022b) | 7-7-7-7 | 47.3 | 65.9 | 51.5 | 41.1 | 63.2 | 44.4 |
| ConvNeXt-T (RepLKNET)* (Ding et al., 2022) | 31-29-27-13 | 47.8 | 66.7 | 52.0 | 41.4 | 63.9 | 44.7 |
| SLaK-T | 51-49-47-13 | **48.4** | **67.2** | **52.5** | **41.8** | **64.4** | **45.2** |
| pre-trained for 300 epochs, finetuned for $3\times$ (36 epochs) | | | | | | | |
| ConvNeXt-T (Liu et al., 2022b) | 7-7-7-7 | 50.4 | 69.1 | 54.8 | 43.7 | 66.5 | 47.3 |
| SLaK-T | 51-49-47-13 | **51.3** | **70.0** | **55.7** | **44.3** | **67.2** | **48.1** |

## 6 ANALYSIS OF SLaK

### 6.1 EFFECTIVE RECEPTIVE FIELD (ERF)

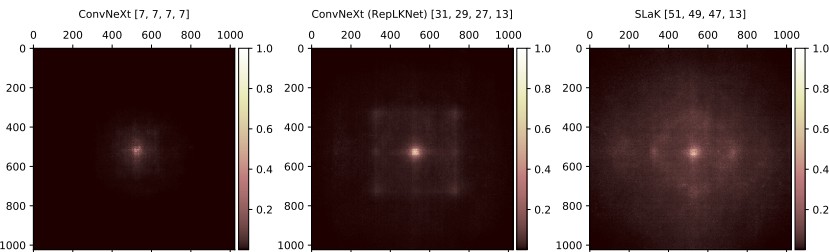

Figure 3: **Effective receptive field (ERF) of models with various kernel sizes**. SLaK is not only able to capture long-range dependence but also the local context features.

The concept of receptive field (Luo et al., 2016; Araujo et al., 2019) is important for deep CNNs: anywhere in an input image outside the receptive field of a unit does not affect its output value. Ding et al. (2022) scale kernels up to $31 \times 31$ and show enlarged ERF and also higher accuracy over small-kernel models (He et al., 2016). As shown in the ERF theory (Luo et al., 2016), ERF is proportion to $\mathcal{O}(k\sqrt{n})$, where $k$ and $n$ refers to the kernel size and the network depth, respectively. Therefore, the hypothesis behind the kernel decomposition in SLaK is that the two decomposed M×N and N×M kernels can well maintain the ability of large kernels in terms of capturing large ERF, while also focusing on fine-grained local features with the shorter edge (N).

To evaluate this hypothesis, we compare the ERFs captured by SLaK and RepLKNet. Following Kim et al. (2021); Ding et al. (2022), we sample and resize 50 images from the validation set to 1024×1024, and measure the contribution of the pixel on input images to the central point of the feature map generated in the last layer. The contribution scores are further accumulated and projected to a 1024×1024 matrix, as visualized in Figure 3. In the left sub-figure, although the original ConvNeXt already improves the kernel size to 7×7, its high-contribution pixels concentrate in the center of the input. Even the 31×31 kernels used by RepLKNet are not sufficient for ConvNeXt to cover the whole input. In comparison, high-contribution pixels of SLaK spread in a much larger ERF, and some

high-contribution pixels emerge in non-center areas. This observation is in line with our hypothesis that SLaK balances between capturing long-range dependencies and focusing on the local details.

## 6.2 KERNEL SCALING EFFICIENCY

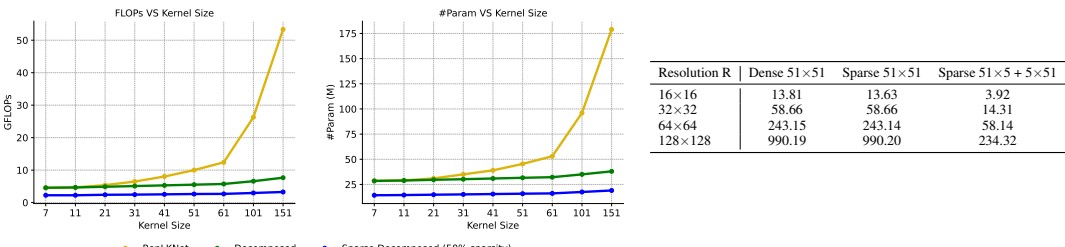

| Resolution R | Dense 51×51 | Sparse 51×51 | Sparse 51×5 + 5×51 |
|---|---|---|---|
| 16×16 | 13.81 | 13.63 | 3.92 |
| 32×32 | 58.66 | 58.66 | 14.31 |
| 64×64 | 243.15 | 243.14 | 58.14 |
| 128×128 | 990.19 | 990.20 | 234.32 |

Figure 4: **Left: Scaling efficiency.** The number of GFLOPs (left) and parameters (right) as the kernel size in ConvNeXt-T scales up. **Right: Real inference time latency (ms).** Real inference time latency of depth-size convolutions with different kernel size. The results are obtained on a single A100 GPU with PyTorch 1.10.0 + cuDNN 8.2.0, in FP32 precision, using no special hardware accelerator.

As Section 4 mentioned, the two components of SLaK, kernel decomposition and sparse group, substantially improve the scaling efficiency of kernel sizes. To further support this, we report the overhead required by various large kernel training methods in Figure 4-left. We simply replace all the kernels in stages of ConvNeXt-T with a set of kernel sizes from 7 to 151 and report the required GFLOPs and the number of parameters. One can clearly see the big gap between full-kernel scaling (yellow lines) and kernel decomposition (green lines) as the kernel size increases beyond 31×31. Even using the ultra-large 151×151 kernels, using our methods would require fewer FLOPs and parameters, compared to full-kernel scaling with 51×51 kernels. Growing evidence has demonstrated that training with high resolution is a performance booster for classification (Tan & Le, 2019; Liu et al., 2021d) and object detection (He et al., 2017; Lin et al., 2017). We believe that large kernels will benefit more in this scenario.

We also report the real inference time latency (ms) of different large-kernel recipes in Figure 4-right using one-layer depth-wise convolutions. The results are obtained on a single A100 GPU with PyTorch 1.10.0 + cuDNN 8.2.0, in FP32 precision, using ***no dedicated sparsity-friendly hardware accelerator***. The input shape is (64, 384, R, R). In general, without special hardware support, while vanilla sparse large kernels cost similar run time to dense large kernels of the same size (sparse kernels receive limited support in common hardware), the sparse decomposed kernels yield more than 4× real inference speed acceleration than directly using vanilla large kernels.

## 7 CONCLUSION

Recent works on modern ConvNets defend the essential roles of convolution in computer vision by designing advanced architectures and plugging large kernels. However, the largest kernel size is limited to 31×31 and the performance starts to saturate as the kernel size continues growing. In this paper, we investigate the training of ConvNets with extremely large kernels that are beyond 31×31 and consequently provide a recipe for applying extremely large kernels inspired by sparsity. Based on this recipe, we build a pure ConvNet model that smoothly scales up the kernel size beyond 51×51, while achieving better performance than Swin Transformer and ConvNeXt. Our strong results suggest that sparsity, as the "old friend" of deep learning, can make a promising tool to boost network scaling.

## ACKNOWLEDGEMENT

S. Liu, X. Chen and Z. Wang are in part supported by the NSF AI Institute for Foundations of Machine Learning (IFML). We thank Zhenyu Zhang for helping conduct many experiments in Section J. Part of this work used the Dutch national e-infrastructure with the support of the SURF Cooperative using grant no. NWO2021.060, EINF-2694 and EINF-2943/L1.

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

**Contents of the Appendices:**

## A    EXPERIMENTAL SETTINGS

### A.1    IMAGENET-1K

We share the (pre-)training settings of SLaK on ImageNet-1K in this section. We train SLaK for 300 epochs (Section 5.1) and 120 epochs (Section 4) using AdamW (Loshchilov & Hutter, 2019) with a batch size of 4096, and a weight decay of 0.05. The only difference between models training for 300 epochs and 120 epochs is the training time. The learning rate is 4e-3 with a 20-epoch linear warmup followed by a cosine decaying schedule. For data augmentation, we use the default setting of RandAugment (Cubuk et al., 2020) in Timm (Wightman, 2019) – "rand-m9-mstd0.5-inc1", Label Smoothing (Szegedy et al., 2016) coefficient of 0.1, Mixup (Zhang et al., 2017) with $\alpha = 0.8$, Cutmix (Yun et al., 2019) with $\alpha = 1.0$, Random Erasing (Zhong et al., 2020) with $p = 0.25$, Stochastic Depth with drop rate of 0.1 for SLaK-T, 0.4 for SLaK-S, and 0.5 for SLaK-B, Layer Scale (Touvron et al., 2021c) of the initial value of 1e-6, and EMA with a decay factor of 0.9999. We train SLaK-T with NVIDIA A100 GPUs and the rest are trained with NVIDIA V100.

### A.2    SEMANTIC SEGMENTATION ON ADE20K

We follow the training setting used in Ding et al. (2022); Liu et al. (2022b) using UperNet (Xiao et al., 2018) implemented by MMSegmentation (Contributors, 2020) with the 80K/160K-iteration training schedule. We conduct experiments with both short and long training procedures. The backbones are pre-trained on ImageNet-1K with 224×224 input for 120/300 epochs and then are fine-tuned with UperNet  (Xiao et al., 2018) for 80K/160K iterations, respectively. We report the mean Intersection over Union (mIoU) with a single scale. All the hyperparameters are exactly the same as the ones used in the official GitHub repository of ConvNeXt (con, 2021).

### A.3    OBJECT DETECTION AND SEGMENTATION ON COCO

For COCO experiments, we follow the training settings used in BEiT, Swin, and ConvNeXt using MMDetection (Chen et al., 2019) and MMSegmentation (Contributors, 2020) toolboxes.  The final model weights are adopted (instead of EMA weights) from ImageNet-1K pre-training with 224×224 input. We also conduct experiments with both short and long training procedures. The backbones are pre-trained on ImageNet-1K with 224×224 input for 120/300 epochs and then are finetuned with Cascade Mask R-CNN (Cai & Vasconcelos, 2018) for 12/36 epochs, respectively. All the hyperparameters are exactly the same as the ones used in the official GitHub repository of ConvNeXt (con, 2021).

## A.4 OBJECT DETECTION ON PASCAL VOC 2007

We follow Liu et al. (2021e) and finetune Faster-RCNN on PASCAL VOC dataset with SLaK-T as the backbone. We use multi-scale setting (Carion et al., 2020; Sun et al., 2021) which leads to the length of the shorter side between 480 and 800 and the ones of the longer side at most 1333. The model is trained with AdamW for 36 epochs with a learning rate of 0.0001, a weight decay of 0.05, and a batch size of 16.

## B DYNAMIC SPARSITY

Dynamic Sparsity (Mocanu et al., 2018; Liu et al., 2021b) is a class of methods that allow for end-to-end training of neural networks with sparse connectivity, also known as sparse training. Dynamic Sparsity starts from a sparse network while jointly optimizing the sparse connectivity and model weights during training. Without loss of generality, we provide the general pseudocode of Dynamic Sparsity that can cover most of the existing methods in Algorithm 1.

While there is an upsurge in increasingly efficient ways for sparse training (Bellec et al., 2018; Mostafa & Wang, 2019; Dettmers & Zettlemoyer, 2019; Evci et al., 2020; Liu et al., 2021c; Jayakumar et al., 2020; Chen et al., 2021; Schwarz et al., 2021; Jiang et al.), most dynamic sparsity techniques share three key components: sparse model initialization, model weight optimization, and sparse weight adaptation. We explain these components in more detail below.

---

**Algorithm 1** Dynamic Sparsity

**Require:** Dense neural network $\boldsymbol{\theta}$, sparse Neural Network $\boldsymbol{\theta}_s$ dataset $\{x_i, y_i\}_{i=1}^{N}$, sparsity $s$, binary masks: $\mathbf{M} = \{\mathbf{M}^1, \ldots, \mathbf{M}^L\}$ where $\{\mathbf{M}^1, \ldots, \mathbf{M}^L\}$ refer to the binary mask from layer 1 to L, Adaptation Frequency $\Delta T$, Adaptation Rate $p$

1:    # Sparse Model Initialization
2:    $\mathbf{M} \leftarrow \text{SNIP}(\boldsymbol{x}; \boldsymbol{\theta}; s)$            ▷ *Using SNIP to generate sparse masks.*
3:    $\boldsymbol{\theta}_s \leftarrow \boldsymbol{\theta} \odot \mathbf{M}$            ▷ *Applying sparse masks.*
4: **for** each training step $t$ **do**
5:      # Model Weight Optimization
6:      $\boldsymbol{\theta}_s \leftarrow \text{AdamW}(\boldsymbol{x}; \boldsymbol{\theta}_s)$
7:      $\boldsymbol{\theta}_s \leftarrow \boldsymbol{\theta}_s \odot \mathbf{M}$            ▷ *Applying masks.*
8:      **if** $(t \mod \Delta T) = 0$ **then**
9:         # Sparse Weight Adaptation
10:        Pruning $p$ percentage of parameters using *magnitude pruning*
11:       Growing $p$ percentage of parameters in a *random* fashion
12:       Update the adaptation rate $p$ with a *cosine decay* schedule
13:      **end if**
14: **end for**

---

## B.1 SPARSE MODEL INITIALIZATION

Starting training from a sparse model is a fundamental requirement of dynamic sparsity (also known as sparse training). Therefore, it is crucial to construct the sparse model in a way that ensures both trainability and capacity are preserved. One critical aspect of constructing a good sparse model is choosing the appropriate layer-wise sparsity ratio. This ratio determines the computational FLOPs (floating-point operations) of the sparse model and has a significant impact on its final performance (Evci et al., 2020; Liu et al., 2022a; Hoang et al., 2023).

Mocanu et al. (2018) first introduced *Erdős-Rényi* (ER) (Erdős & Rényi, 1959) from graph theory to the field of neural networks, achieving better performance than the standard uniform sparsity ratio. Evci et al. (2020) further extended ER to CNN and brings significant gains to sparse CNN training with the *Erdős-Rényi-Kernel* (ERK) ratio. Other prior arts start from a uniform sparsity distribution while allowing the distribution dynamically shift towards a non-uniform one during training. Liu et al. (2022a) recently discover that the layer-wise sparsity ratio learned by SNIP (Lee et al., 2019) outperforms ER-based sparsity ratios with large-scale models on ImageNet. We confirm that SNIP ratio indeed achieves a better trade-off between accuracy and FLOPs than ERK in the

context of large-scale ConvNets. Concretely, we first calculate the SNIP scores $|g \odot w|$ for each weight using one batch of training data, where $w$ and $g$ is the network weight and gradient, respectively. Then, we globally sort all the scores across layers and force the weights with the smallest scores to be zero with respect to the target sparsity.

## B.2  Model Weight Optimization

After initializing the sparse model, we start to optimize the model weights to minimize the loss function. Dynamic sparsity is compatible with the most widely used optimizers. In this paper, we directly inherit the default training configurations and optimizers (AdamW) from ConvNeXt (Liu et al., 2022b) without any modifications due to its already promising performance. However, we do not exclude the possibility to introduce sparsity-specific components such as optimizers, normalizations, and activation functions to further enhance the performance. After each update, we multiply the weights with the current binary masks to enforce the sparse structure.

## B.3  Sparse Weight Adaptation

Sparse weight adaptation is the vital factor for dynamic sparsity to outperform static sparsity (Liu et al., 2021c). By dynamically adapting the sparse weights during training, it enables joint optimization of sparse connections and weights for a more elaborate capture of local features. Prune-and-grow is the most common way to achieve weight adaptation. Removing $p$ percentage of the existing weights and regrowing the same number of new weights explores new weights while is able to maintain the parameter count fixed.

**Weight Prune:** Although there exists various pruning criteria in pruning literature, we choose the most common way for SLaK, that is, removing the weights with the smallest magnitude (Mocanu et al., 2018).

**Weight Grow:** The most common ways to grow new weights are random-based growth (Mocanu et al., 2018) and gradient-based growth (Dettmers & Zettlemoyer, 2019; Evci et al., 2020). We choose to grow new weights by randomly sampling following (Mocanu et al., 2018).

Following Liu et al. (2021c), we specifically tune two factors for SLaK-T that control the strength of weight adaptation, adaptation frequency $f$, and adaptation rate $p$. Adaptation frequency determines after how many training iterations we adjust the sparse weights, and the latter controls the ratio of the weight that we adjust at each adaptation. We share the results in Figure 5. We empirically find that $f = 100$ and $p = 0.3$ works best for SLak-T and choose the same set of hyperparameters for SLak-S/B, without further tuning.

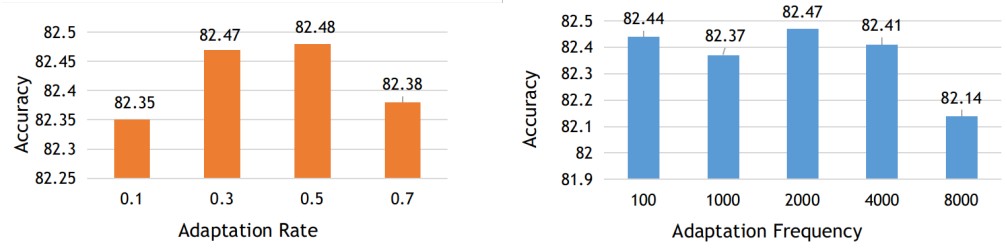

Figure 5: **Left:** Effect of adaptation rate $p$ on the performance of SLaK-T. $f$ is set as 100. **Right:** Effect of the adaptation frequency $f$ on the performance of SLaK-T. $p$ is set as 0.3.

## C  Comparison between CSwin Transformer and SLaK

We examine if SLaK can rival CSwin Transformer (Dong et al., 2022), a carefully designed hybrid architecture with transformers and convolutions. It performs self-attention in both horizontal and vertical stripes. To produce a hierarchical representation, convolution layers are also used in CSwin Transformer to expand the channel dimension. As shown in Table 7, besides clearly outperforming

the peer ConvNets backbones (ConvNeXt and RepLKNet), SLaK models can also perform on par with CSwin Transformers, as a pure ConvNet for the first time, without any bells and whistles.

Table 7: **Comparison with the CSwin Transformer.** All models are trained with AdamW for 300 epochs.

| Model | Image Size | #Param. | FLOPs | Top-1 Accuracy (%) |
|---|---|---|---|---|
| CSWin-T (Dong et al., 2022) | 224×224 | 23M | 4.3G | **82.7** |
| CSWin-S (Dong et al., 2022) | 224×224 | 35M | 6.9G | 83.6 |
| CSWin-B (Dong et al., 2022) | 224×224 | 78M | 15.0G | **84.2** |
| CSWin-B (Dong et al., 2022) | 384×384 | 78M | 47.0G | 85.4 |
| **SLaK-T** | 224×224 | 30M | 5.0G | 82.5 |
| **SLaK-S** | 224×224 | 55M | 9.8G | **83.8** |
| **SLaK-B** | 224×224 | 95M | 17.1G | 84.0 |
| **SLaK-B** | 384×384 | 95M | 50.3G | **85.5** |

## D  TRADE-OFF BETWEEN SPARSITY AND WIDTH

The principle of "use *sparse* groups, expand more" in Observation #3 essentially sacrifices network sparsity for network width. Therefore, there is a trade-off between model sparsity and width. To have a better understanding of this trade-off, we choose 5 combinations of (Sparsity, Width factor), i.e., $(0.20, 1.1\times)$, $(0.40, 1.3\times)$, $(0.55, 1.5\times)$, $(0.70, 1.9\times)$ and $(0.82, 2.5\times)$. All the settings have roughly 5.0M FLOPs but with different network widths. The experiments are conducted on SLaK-T. As we expected, the model's performance keeps increasing as model width increases until the width factor reaches $1.5\times$, after which increasing width further starts to hurt the performance apparently due to the training difficulties associated with highly sparse neural networks.

Table 8: **Trade-off between sparsity and width.** The experiments are conducted on SLaK-T trained for 120 epochs on ImageNet-1K.

| Model | (Sparsity, Width) | #Param | FLOPs | Top-1 Accuracy (%) |
|---|---|---|---|---|
| | $(0.20, 1.1\times)$ | 29M | 5.0G | 81.3 |
| | $(0.40, 1.3\times)$ | 30M | 5.0G | 81.6 |
| SLaK-S | $(0.55, 1.5\times)$ | 30M | 4.9G | **81.7** |
| | $(0.70, 1.9\times)$ | 32M | 5.0G | 81.6 |
| | $(0.82, 2.5\times)$ | 32M | 5.1G | 81.3 |

## E  EFFECT OF THE SHORTER EDGE N ON SLaK

In Table 9, we report the effect of the shorter edge on the performance of SLaK. We vary the shorter edge $N \in [3, 5, 7]$ and report the accuracy. All models were trained with AdamW on ImageNet-1K for 120 epochs. We empirically find that N=5 give us the best results, whereas $N = 3$ and $N = 7$ has slightly lower accuracy. We hence choose $N = 5$ as our default option.

Table 9: **Effect of the shorter edge N on SLaK.** The experiments are conducted on SLaK-T trained for 120 epochs on ImageNet-1K.

| Model | N | Top-1 Accuracy (%) |
|---|---|---|
| SLaK-T | 3 | 81.5 |
| SLaK-T | 5 | 81.6 |
| SLaK-T | 7 | 81.4 |

Table 10: **Quantitative analysis on the ERF with the high-contribution area ratio** $r$. A larger value suggests a smoother distribution of high-contribution pixels, hence larger ERF.

|  | Kernel Size | $t = 20\%$ | $t = 30\%$ | $t = 50\%$ | $t = 99\%$ |
|---|---|---|---|---|---|
| ResNet-101 | 3-3-3-3 | 0.9% | 1.5% | 3.2% | 22.4% |
| ResNet-152 | 3-3-3-3 | 1.1% | 1.8% | 3.9% | 34.4% |
| ConvNeXt-T | 7-7-7-7 | 2.0% | 3.6% | 7.7% | 55.5% |
| ConvNeXt-T (RepLKNet) | 31-29-27-13 | 4.0% | 9.1% | 19.1% | 97.5% |
| SLaK-T | 51-49-47-13 | **6.9%** | **11.5%** | **23.4%** | **97.5%** |

## F  ERF QUANTITATION OF MODELS WITH DIFFERENT KERNEL SIZES

In this appendix, we quantify ERF of various models by reporting the high-contribution area ratio $r$ in Table 10 following Ding et al. (2022). $r$ refers to the ratio of a minimum rectangle to the overall input area that can cover the contribution scores over a given threshold $t$. Assuming an area of A×A at the center can cover $t = 30\%$ contribution scores of a 1024×1024 input, then the area ratio of $t = 30\%$ is $r = (A/1024)^2$. Larger $r$ suggests a smoother distribution of high-contribution pixels. We can see that with global kernels, SLaK naturally considers a larger range of pixels to make decisions than ConvNeXt and RepLKNet.

## G  LEARNING CURVE

We here share the learning curve of different architectures on ImageNet-1K classification in Figure 6. Again, we can observe that large kernels provide significant training loss gains (yellow lines and blue lines). Increasing kernel size from 31×31 to 51×51 further decreases the training loss with improved test accuracy. It is worth noting that even though SLaK's kernel size is as large as 51×51, it enjoys a very promising convergence speed compared to the models with smaller kernel sizes.

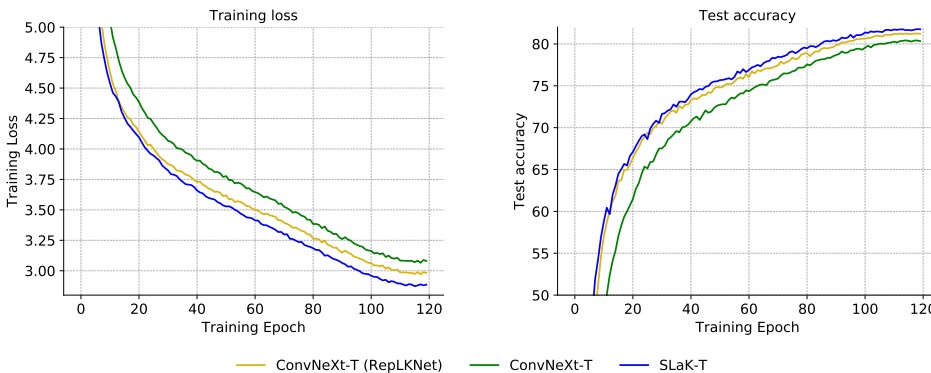

Figure 6: **Learning Curve.** Learning curves in terms of training loss and test accuracy of various models on ImageNet-1K classification.

## H  INFERENCE THROUGHPUT MEASUREMENT

Although our paper shows promising benefits of dynamic sparsity, we are unable to fully translate these benefits into real speedup due to the limited support of unstructured sparsity in commonly used hardware without sparsity-aware accelerations.

To provide a full picture of this aspect, we measure the real inference throughput of SLaK-T, Swin-T, and ConvNeXt-T with an A100 GPU with PyTorch 1.10.0 + cuDNN 8.2.0, in FP32 precision. The results are reported in Table 11. On common GPUs without any sparsity-aware accelerators, the throughput of Swin-T and ConvNeXt-T is around 1.8× and 2.5× larger than SLaK-T. This difference

is acceptable, considering SLaK's $1.3\times$ wider model size and $>7\times$ larger kernel size, thanks to the kernel decomposition trick. Naively scaling the kernel size of ConvNeXt to $51\times51$ without kernel decomposition will result in an $8.5\times$ slowdown compared to the original $7\times7$ kernel.

Table 11: **Inference throughput comparisons among ConvNeXt-T, Swin-T, and SLaK-T on ImageNet-1K.** The results are obtained on an A100 GPU with PyTorch 1.10.0 + cuDNN 8.2.0, in FP32 precision, using no sparsity-aware accelerator.

| Models | Image Size | Kernel Size | Throughput ↑ |
|---|---|---|---|
| ConvNeXt-T | 224×224 | 7-7-7-7 | 1779 |
| Swin-T | 224×224 | - | 1305 |
| SLaK-T | 224×224 | 51-19-17-13 | 709 |

To better understand the effect of different components of our training recipe on real inference speed (without sparsity-aware accelerators), we break down our training recipe and report the corresponding inference throughput in Table 12. Generally speaking, sparse large kernels have comparable run times to dense large kernels, with neither acceleration nor overhead, as the masking operation of sparse kernels is essentially equivalent to dense matrix multiplication, which is highly efficient on GPUs. However, decomposing large kernels can significantly increase the efficiency of kernel scaling, achieving a $3\times$ speedup compared to naive kernel scaling. Also due to this reason, the throughput of our $1.3\times$ wider model is still $2\times$ larger than a vanilla large kernel model.

Table 12: **Inference throughput of different large-kernel recipes on ImageNet-1K.** The results are obtained on an A100 GPU with PyTorch 1.10.0 + cuDNN 8.2.0, in FP32 precision, using no sparsity-aware accelerator.

| Model | Kernel Size | Naive | Decomposed | Sparse Groups | Sparse Groups + 1.3× width |
|---|---|---|---|---|---|
| ConvNeXt-B | 51-49-47-13 | 96 | 283 | 282 | 198 |

# I    COMPARISONS WITH OTHER CHOICES FOR LARGE-KERNEL DECOMPOSITION

In this appendix section, we study the effectiveness of our large-kernel decomposition by comparing the following four large-kernel choices:

- Naively stacking 10 small kernels with $3\times3$ size.
- Sequential convolutions with $51\times5$ and $5\times51$ kernels.
- Parallel convolutions with $51\times5$ and $5\times51$ kernels (same as SLaK).
- Dilation convolutions with a similar receptive field to SLaK, i.e., we set the kernel size of each stage as [19, 17, 17, 5] with a dilation factor of 3.

To eliminate the effect of confounding variables, dynamic sparsity is not adopted here and an additional $5\times5$ kernel is added for all settings. We can observe from Table 13 that naively stacking multiple small kernels and dilation kernels achieve much lower accuracy than the decomposed approaches (parallel and sequential). And the sequential decomposition performs slightly worse than the parallel one by 0.1%. The superior performance of kernel decomposition (either sequential or parallel) over other choices also supports the effectiveness of our recipe.

Table 13: **Comparisons with other choices for large-kernel decomposition.** All models are trained for 120 epochs.

| Model | Stacking 10 of 3×3 kernels | Dilation Convolutions | Sequential 51×5 + 5×51 | Parallel 51×5 + 5×51 |
|---|---|---|---|---|
| ConvNeXt-T | 80.6 | 81.1 | 81.4 | 81.5 |

By combining the 51×5 and 5×51 kernels through superposition, we obtain a restricted 51×51 kernel that is primarily composed of weights arranged in a crossed shape. This restricted filter offers several advantages, including computational efficiency, improved locality, and most importantly, the large effective receptive field (ERF) similar to a 51×51 kernel. The kernel decomposition idea is related to prior work on self-attention decomposition in Transformers (Ho et al., 2019; Tu et al., 2022). In those models, a global self-attention mechanism is replaced with a combination of local attention and sparse global attention, or with axial attention that performs attention along different image axes. These approaches aim to achieve computational efficiency while preserving the global ERF. Our empirical results provide evidence that a similar decomposed structure can be extended as a useful design principle for ConvNets too.

## J    EFFECT OF LARGE KERNEL POSITION

In the main body of our paper, we follow the design of RepLKNet (Ding et al., 2022) and respectively scale the kernel size from [31, 29, 27, 13] to [51, 47, 47, 13] for each model stage without careful design. In this section, we question the necessity and rationality of using large kernels at every stage. For instance, since early stages usually learn low-level features such as edges, corners, basic local shape information, large kernels like 31×31 or 51×51 might not be necessary in early stages. To verify this, we conduct an ablation study by naively scaling large kernels only in one or two stages. Dynamic sparsity and decomposition are not adopted to eliminate the effect of confounding variables.

The results are presented in Table 14. As we anticipated, solely scaling the kernel size to 31×31 in the first and or 29×29 in the second stage does not yield any benefits. Instead, large kernels are more crucial at later stages. Only increasing the kernel size in the third and forth stage achieve the same accuracy as when scaling kernels in all stages. Our results indicate that scaling the kernel sizes in the later stages is a better design option for efficiency and accuracy.

Table 14: **Ablation study of the position of large kernels.** All models are trained for 120 epochs. [A, B, C, D] refers to the kernel size at each model stage.

| Kernel Size | 7-7-7-7 | 31-7-7-7 | 7-29-7-7 | 7-7-27-7 | 7-7-7-13 | 7-7-27-13 | 31-29-27-13 |
|---|---|---|---|---|---|---|---|
| ConvNeXt-T | 81.0 | 80.89 | 80.94 | 81.31 | 81.18 | 81.48 | 81.5 |

## K    LIMITATIONS AND DISCUSSION OF BROADER IMPACT

The main limitation of this work is that the sparse architecture is implemented with binary masks due to the limited support of sparse neural networks by the common hardware such as GPU and TPU. Therefore, the inference FLOPs reported in the main paper are the theoretical values. Traditional works on structured sparsity (Han et al., 2015; Frankle & Carbin, 2019; Gale et al., 2020; Liu et al., 2021a) mainly focus on finding sparse subnetworks that can match the performance of their dense counterparts. The promising results in our paper go one step further and demonstrate the large potential of unstructured sparsity for scaling modern neural architectures. Once this great potential is supported in the future, it can have a significant positive impact on our planet by saving a huge amount of energy and reducing overall total carbon emissions. Although not the focus of this current work, it would be interesting for future work to examine the speedup of sparse large kernels, using such specialized hardware accelerators, as we see much improvement room of promise here. For example, at high unstructured sparsity levels, XNNPACK (Elsen et al., 2020) has already shown significant speedups over dense baselines on smartphone processors. Furthermore, an increasing number of companies and researchers are considering including unstructured sparsity support in their hardware (Liu et al., 2021b; Gale et al., 2020; Nvidia, 2020; Zhou et al., 2021; Hubara et al., 2021). We hope our work can provide more motivation for such advances.

While our paper is solely scientific, others could build software with a negative impact using our technique, like privacy leakage and fairness issues. Moreover, the promising results of sparsity could bring benefits as cost savings, but could also possibly result in a debate between dense NN hardware and sparse NN hardware, which might lead to resource waste.

