# OpenReview forum: "More ConvNets in the 2020s: Scaling up Kernels Beyond 51x51 using Sparsity"
_ICLR.cc/2023/Conference — ICLR 2023 poster_

### Official Review · Reviewer_h6xx · 2022-10-21

**Confidence:** 4
**Correctness:** 4
**Technical Novelty And Significance:** 2
**Empirical Novelty And Significance:** 2
**Recommendation:** 6

**Clarity, Quality, Novelty And Reproducibility:**

The paper is clear in its description and provides necessary information for the experimental setup. It could do a better job at clarifying for the reader what message they should walk away with.

My biggest concern is the novelty, as the building blocks used are (if I understand correctly) known, and their combination is natural based on the existing art. This would not be a concern if the paper demonstrated that the particular combination is the thing practitioners should do going forward, but for that I would want to see a broader and more thorough empirical analysis (comparing to more architectures, on different datasets, tasks, and perhaps even domains).

**Strength And Weaknesses:**

The paper proposes a simple way for scaling up the size of the convolutional kernels, and empirically demonstrates the effectiveness of this  on several visual tasks. The kernels can be scaled to a larger size than before, while boosting the performance.

The paper proposes a mechanism for introducing dynamic sparsity, which improves the performance at no extra cost in FLOPS.

The experimental results explore different training schedules, different tasks (classification, segmentation, detection), and (positively) compare to several SOTA architectures.

It was not clear to me what the main message of the paper is, other than that it is possible to train CNNs with large kernels. For example, is it the case that the best practice is to use such models? If so, I would have liked to see experiments on more data and tasks, including ones with large-data pretraining and possibly transfer learning (as with e.g. the VTAB suite of benchmarks). Comparing to more architectures, with results perhaps presented as a pareto frontier in the cost / performance plane, would be instructive.

If the main message is the fact that sparsity is important, this should be combined with the analysis of the actual cost of sparse inference, which (as the paper mentions) cannot be measured just using FLOPS.

If the message is that a larger receptive field could be achieved by a parallel combination of non-square filters, this seems known (since at least the Inception models). I should also note that while the paper refers to 51x51 filters, it is my understanding that it considers a restricted form of such filters, which are cross-shaped. It would be useful to understand why such a shape is a good one to use -- as opposed to e.g. convolving with the two filters sequentially rather than in parallel which would give something closer to a square filter, or atrous convolutions which provide a sparse coverage of a large receptive field.

I was looking forward to understanding why previous work did not use larger kernels, and the first sentence of sec. 3 seemed to promise 3 reasons for it, which I did not find.



**Summary Of The Paper:**

The paper proposes a mechanism for scaling up the receptive field of the convolutional kernels in CNNs, and for improving the performance without extra FLOPS via sparsity. Positive results are shown for several visual tasks.

**Summary Of The Review:**

The method is simple and results are encouraging. However, I have concerns with novelty. Even if the method is a natural combination of known components, the paper could clarify / analyze why the specific set of choices is the "right" one to use, and provide more complete empirical validation.

~In the current form, I believe the paper falls short for ICLR.~

*Edit 11/28:* During the discussion phase, we have identified ways for the paper to be edited to call attention to the salient aspects, specifically one around the cross-shaped filters allowing a beneficial trade-off between the parameter efficiency and the receptive field size. Raising the score from 5 to 6.

---

> ### Author Response · Authors · 2022-11-15
> **Response to Reviewer h6xx (2/2)**
>
> **Comment2: If the message is that a larger receptive field could be achieved by a parallel combination of non-square filters, this seems known (since at least the Inception models). I should also note that while the paper refers to 51x51 filters, it is my understanding that it considers a restricted form of such filters, which are cross-shaped. It would be useful to understand why such a shape is a good one to use -- as opposed to e.g. convolving with the two filters sequentially rather than in parallel which would give something closer to a square filter, or atrous convolutions which provide a sparse coverage of a large receptive field.**
>
> * While kernel decomposition was introduced in Inception families, the possibility of using kernel decomposition to scale up kernel size up to 51$\times$51 or 61$\times$61 has never been explored. We for the first time unveil that using kernel decomposition can surprisingly reverse the accuracy descending trend of scaling kernels beyond 31$\times$31, i.e., the decomposed 51$\times$51 kernels achieve better performance than 31$\times$31.
>
> * To further study the effectiveness of our recipe, we added three baselines as you suggested: (1) naively stacking 3 3$\times$3 kernels instead of a combination of 51$\times$5 and 5$\times$51; (2) sequentially convolving with 51$\times$5 and 5$\times$51 kernels; (3) dilation/atrous convolutions with a similar receptive field to SLaK, i.e., we set the kernel size of each stage as [19, 17, 17, 5] and a dilation factor of 3. To eliminate the effect of other components, we remove the dynamic sparsity and keep a parallel 5$\times$5 kernel to all of these baselines. We can observe that naively stacking multiple small kernels and dilation kernels in general achieve much lower accuracy than the decomposed 51$\times$51 methods. And the sequential decomposition slightly underperforms the parallel one by 0.1%. The superior performance of kernel decomposition (either sequential or parallel) over other choices also supports the effectiveness of our recipe. We hence think it reasonable to stick to our parallel kernel decomposition as the default option.
> | Model  | 3 stacked 3$\times$3 | dilation conv | sequential 51$\times$5 and 5$\times$51 |parallel 51$\times$5 and 5$\times$51|
> | ------------- | :-----------: |:-----------:|:-----------: |:-----------:|
> | SLaK-T | 80.2  | 81.1 | 81.4  | 81.5 |
>
> **Comment3: I was looking forward to understanding why previous work did not use larger kernels, and the first sentence of sec. 3 seemed to promise 3 reasons for it, which I did not find.**
> * The three observations mentioned in the first sentence of Section 3 is not about the reasons why previous work did not use larger kernels, but three observations that we summarized from our large kernel study in Section 3 and Section 4: (1) Previous methods fail to scale kernels beyond 31x31; (2) Decomposing a large kernel into two rectangular, parallel kernels smoothly scales the kernel
> size up to 61×61; (3) “Use sparse groups, expand more width” significantly boosts the model capacity. We have remove this sentence in our revision to avoid any potential misleading.

---

> ### Author Response · Authors · 2022-11-15
> **Response to Reviewer h6xx (1/2)**
>
> We thank you very much for your time and review. We provide detailed feedback to address your comments below.
>
> **Comment1: My biggest concern is the novelty, as the building blocks used are (if I understand correctly) known, and their combination is natural based on the existing art.  It was not clear to me what the main message of the paper is, other than that it is possible to train CNNs with large kernels. For example, is it the case that the best practice is to use such models? If so, I would have liked to see experiments on more data and tasks, including ones with large-data pretraining and possibly transfer learning (as with e.g. the VTAB suite of benchmarks). Comparing to more architectures, with results perhaps presented as a pareto frontier in the cost/performance plane, would be instructive. If the main message is the fact that sparsity is important, this should be combined with the analysis of the actual cost of sparse inference, which (as the paper mentions) cannot be measured just using FLOPS.**
>
> * First, please allow us to re-emphasize that our goal is not to claim the best model to be used in practice, but to discover scientifically how much potential improvement space that pure ConvNets remain to have by purely enlarging kernel size, in the era of Transformers. Following the recent endeavors of ConvNeXt and RepLKNet that draw explosive interest from the community, we push along the direction of pure ConvNets and for the first time scaled kernel size up to 51$\times$51 or even 61$\times$61, which has never been achieved before. The fundamental contribution of our paper is to validate the potential of continually scaling up convolutional kernels to improve SOTA performance, which is highly non-trivial as seen from our observations (e.g., the vanilla scaling of kernels beyond 31$\times$31 using existing techniques will start to hurt the performance). Hence, each individual component of our recipe is contributing to the novel goal and unified recipe for scaling kernel size up to an unprecedented level: 51$\times$51. With extensive experiments on ImageNet classification and 3 dense prediction tasks, we show that our models with 51$\times$51 consistently achieve better performance than previous ConvNets, while being better than or on par with SOTA ViTs.
>
> * The novelty and merits of our paper have also been well recognized by other reviewers with positive comments like **Reviewer rrkL**: "Both empirical results and scientific insights appear to be strong", "a highly unusual architecture", "scientifically meaningful", "Novelty is overall good bold"; **Reviewer X26i**: "a novel convolutional network", "the first work that scales kernel size to 51x51", "The idea is very novel", "strong novelty"; **Reviewer YfwT**: "To the best of my knowledge the proposed approach is new", etc.

---

> ### Author Response · Authors · 2022-11-26
> **Gentle Reminder**
>
> Dear Reviewer **h6xx**,
>
> Thank you again for your time reviewing our paper and your helpful comments! We would appreciate it if you could confirm that our responses address your concerns. We would also be happy to engage in further discussions to address any other questions that you might have.
>
> Best regards,
>
> Authors

---

### Official Review · Reviewer_YfwT · 2022-10-24

**Confidence:** 4
**Correctness:** 3
**Technical Novelty And Significance:** 4
**Empirical Novelty And Significance:** 4
**Recommendation:** 8

**Clarity, Quality, Novelty And Reproducibility:**

To the best of my knowledge the proposed approach is new. The paper is clearly written, and re-implementation of the idea and experiments should be possible from just the paper alone.


**Strength And Weaknesses:**

### Strengths:

- Clear idea nicely explained.
- Good evaluation, comparing against a number of alternative state of the art models.

### Weaknesses:

- Some of the results might arguable lie within the bounds of error - whilst performaing a sensitivity analysis might not be possible, could the authors say something about learnability of the models? are they easier or more difficult to train than the alternatives?
- The last paragraph of section 6.1 is probably going a bit far - whilst it is true that human vision is foveated, the way that a human looks at an image is somewhat different to how a CNN percieves it. (I'd personally just tone down the links to human vision here)
- The statement on negative societal impacts is very naive and should be either removed or reconsidered (whilst I agree that this work is scientific, others will undoubtedly build on it and could build software with a negative impact using this technique - you should be open about this even though it is out of your control)


**Summary Of The Paper:**

This paper explores the use of large convolution kernels in CNNs, and demonstrates they can match of outperform transformer based architectures if appropriate allowances are made to reduce the parameter count (by factorisation and sparsity induction). Experiments on ImageNet classification, Semantic segmentation (ADE20K, COCO) and object detection (VOC2007, COCO) show the power of the proposed approach.

**Summary Of The Review:**

This work provides a new approach to making large-kernel convolutions in CNNs realisable. The paper is well written on the whole and the evaluation is good.

---

> ### Author Response · Authors · 2022-11-15
> **Response to Reviewer YfwT**
>
> We sincerely thank you for your review and support! Your support means a lot to us. We provide our response to your constructive comments below.
>
> **Comment 1: Some of the results might arguable lie within the bounds of error - whilst performaing a sensitivity analysis might not be possible, could the authors say something about learnability of the models? are they easier or more difficult to train than the alternatives?**
>
> * We have tried different random seeds and batch sizes and found our models are pretty robust. Moreover, our training recipe is also insensitive to hyperparameters and architectures. For instance, we conduct a lightweight sweep and find that the combination of 1.3$\times$ width and 40% sparsity works the best for SLaK-T. We then directly apply the same set of hyperparameters to SLaK-S and SLaK-B with no ad-hoc tuning, which also provides us with appealing performance improvement over ConvNeXt.
>
> **Comment 2: The last paragraph of section 6.1 is probably going a bit far - whilst it is true that human vision is foveated, the way that a human looks at an image is somewhat different to how a CNN percieves it. (I'd personally just tone down the links to human vision here).**
>
> * Thanks! We agree with you and have removed this connection in our revision to avoid any unwanted confusion.
>
> **Comment 3: The statement on negative societal impacts is very naive and should be either removed or reconsidered (whilst I agree that this work is scientific, others will undoubtedly build on it and could build software with a negative impact using this technique - you should be open about this even though it is out of your control.**
>
> * Thanks for such a constructive comment. We agree with you that even though our paper is solely scientific, others could build software with a negative impact using our technique, like privacy leakage and fairness issues. The promising results of sparsity could bring benefits such as cost savings, but could also result in a debate between dense NN hardware and sparse NN hardware, leading to a potential waste of resources. We have removed our original statement and added the above discussions in Appendix J.

---

> > ### Comment · Reviewer_YfwT · 2022-11-28
> > **thanks for the reply**
> >
> > Just leaving a note to say thank you for the response. The changes will make the paper stronger. Having read the other reviews, I disagree with some of the claims of marginal significance of the novelty results/approach. Clearly this is subjective, but I think you as authors have done enough here. I'm minded to keep my original score as I still think it is overall appropriate for the work presented, even with the changes made from the other reviewers.

---

> > > ### Author Response · Authors · 2022-11-29
> > > **Thank you so much for your suppoert!**
> > >
> > > Dear Reviewer YfwT,
> > >
> > > We sincerely appreciate all your helpful comments and support, which indeed makes our paper stronger.
> > >
> > > We are so glad that you find our novelty significant and our presentation appropriate.
> > >
> > > Best wishes,
> > >
> > > Authors

---

### Official Review · Reviewer_X26i · 2022-10-25

**Confidence:** 5
**Correctness:** 4
**Technical Novelty And Significance:** 4
**Empirical Novelty And Significance:** 3
**Recommendation:** 10

**Clarity, Quality, Novelty And Reproducibility:**

* The paper is well written and easy to follow.
* The idea is very novel, with impressive SOTA results on multiple benchmark datasets.
* The authors provided suffcient materials to reproduce the results.

**Strength And Weaknesses:**

[Strength]

* While using a large kernel size is not a novel idea, it is still the first work that scales kernel size to 51x51 without training difficulty. The most interesting design is the 5x5 small kerenl to capture locality.

* The large-scale experiments show very promising results comparing to ViT-based models. Especially, it outperforms Swin-B by 1% at 384x384 resolution, only using convolutional operators.

* Extensive experiments on ImageNet classification as well as multiple downstream tasks. All showing impressive improvements over baseline methods.

* It is interesting that the rank-5 decomposition of the large kernel is chosen, instead of mx1+1xm. Also, the dynamic training is shown to improve the final accuracy while reducing the model size.

[Weakness]

Although this work already did lots of experiments, I would suggest a few more explorations to enlarge the impact of this work:

* Can we use the proposed model in NLP problems?

* What about stacking multiple 3x3 conv instead of using mx5+5mx+5x5 kernel?

**Summary Of The Paper:**

The authors propose a novel convolutional network structure with very large kernel size, up to 51x51 or even larger. The key ideas is to replace the convolutional kerenl in ConvNeXt by three kernels of size mx5 / 5xm / 5xm respectively. The mx5 + 5xm provides a low-rank approximation to mxm kernel, which can be too large to be learned efficiently. The 5x5 kernel captures the local dependency more efficienctly. During the training, dynamic sparsity is used to adaptive prune-and-grow the kernel weights. Large-scale experiments on ImageNet show that the proposed network can outperform ViT models under the same params and FLOPs.

**Summary Of The Review:**

This is a very strong work, with strong novelty and SOTA results.

---

> ### Author Response · Authors · 2022-11-15
> **Response to Reviewer X26i**
>
> Thank you so much for your time and effort in reviewing our paper! We are significantly encouraged by your positive feedback with high confidence.
>
> **Comment 1: Can we use the proposed model in NLP problems?**
> * Thank you for such an insightful question. Given the extremely large kernel we use, it is interesting to probe the possibility of SLaK in NLP tasks due to its Long-Range Dependency. Some previous arts [1,2] have shown that CNN can achieve comparable performance to RNN on a broad array of NLP tasks including sentiment/relation classification, textual entailment, and answer selection, etc. More interestingly, a very recent work [3] proposes Structured Global Convolution (SGConv), an efficient global convolution whose  the number of parameters scales sub-linearly with sequence length, can achieve strong empirical performance on several NLP tasks such as Long Range Arena and Speech Command.  It would be very interesting to evaluate SLaK on these tasks, since we share a similar property of the Long-Range Dependency. However, we would like to do it in the future as it is almost impossible to do it within this two-week rebuttal window.
>
> **Comment 2: What about stacking multiple 3x3 conv instead of using mx5+5mx+5x5 kernel?**
> * Compared with using large kernels, stacking small 3$\times$3 kernels can be a less efficient way to obtain a large effective receptive field. [4] reports that a 24-layer RepLKNet equipped with 31$\times$31 kernels obtains a much larger effective receptive field than a 152-layer ResNet with 3$\times$3 kernels.
> | Model  | 3 stacked 3$\times$3 kernels | parallel 51$\times$5 and 5$\times$51 (SLaK) |
> | ------------- | :-----------: |:-----------:|
> | SLaK-T | 80.2  | 81.5 |
>
> * To evaluate the performance of stacking multiple 3x3 convolutions, we directly replace the combination of 51$\times$5 and 5$\times$51 with 3 stacked 3$\times$3 kernels, while keeping the 5$\times$5 kernel parallel to them. To make an apple-to-apple comparison, the dynamic sparsity trick is removed for both of them. The following table shows that stacking multiple 3$\times$3 kernels achieves a much lower accuracy than our 51$\times$5 + 5$\times$51 kernels.
>
> [1] [Yin, Wenpeng, et al. "Comparative study of CNN and RNN for natural language processing." arXiv preprint arXiv:1702.01923 (2017).](https://arxiv.org/pdf/1702.01923.pdf)
>
> [2] [Bai, Shaojie, J. Zico Kolter, and Vladlen Koltun. "An empirical evaluation of generic convolutional and recurrent networks for sequence modeling." arXiv preprint arXiv:1803.01271 (2018).](https://arxiv.org/abs/1803.01271)
>
> [3] [Li, Yuhong, et al. "What Makes Convolutional Models Great on Long Sequence Modeling?." arXiv preprint arXiv:2210.09298 (2022).](https://arxiv.org/abs/2210.09298)
>
> [4] [Ding, Xiaohan, et al. "Scaling up your kernels to 31x31: Revisiting large kernel design in cans." Proceedings of the IEEE/CVF Conference on Computer Vision and Pattern Recognition. 2022.](https://arxiv.org/abs/2203.06717)

---

> > ### Comment · Reviewer_X26i · 2022-11-29
> > **Thank you for the author feedback**
> >
> > I do not have any more concerns. A good work with impressive results.

---

### Official Review · Reviewer_rrkL · 2022-10-25

**Confidence:** 3
**Correctness:** 3
**Technical Novelty And Significance:** 3
**Empirical Novelty And Significance:** 3
**Recommendation:** 6

**Clarity, Quality, Novelty And Reproducibility:**

This paper is very well written, clear and concise.  Novelty is overall good bold despite some limitations. Codes are available and results are reproducible to my best knowledge.

**Strength And Weaknesses:**

This is a strong empirical paper studying a highly unusual architecture. This paper did an excellent job in solidifying their approach, validating a large variety of datasets/tasks, and delivering the performance that one expects. I am hence positive overall. The main strengths of this paper are enlisted as follows:
- The problem considered is very interesting: how much gain one could squeeze from ConvNets, by purely enlarging this kernel size. Provided with the recent trend of large-kernel CNNs, scaling kernel size up to 51x51 or even 61x61 was still rarely discussed before.

- Asking this question is also scientifically meaningful: while ViTs seem to dominate vision now with superior performance, it is unclear which truly matters: the larger/global receptive field or the sophisticated adaptive attention weights per location. The observation presented in this paper, that a ConvNet (which sticks to translation-invariant kernels) with (unprecedently) large kernel size outperforms best ViTs, provides a much clearer answer to this understanding.

- The authors show that naively enlarging kernel sizes won’t work, and introduce two tricks: two-way factorization, and dynamic sparsity with more width – the second one is more non-trivial and leverages the recent progress in dynamic sparse training. Their recipe seems very concise and compact, avoiding any ad-hoc architecture modification, and seemingly insensitive to hyperparameters too. Tables 1 and 2 demonstrate whether adding each step of receipt would work or not alone, so that clarifies which part contributes how much to the final performance.

- The performance gains are significant and consistent on ImageNet, surpassing some of the best available models (SwinT, ConvNext, RepLKNet) across four scales (T, S, B, and B with high-resolution 384 input). It performs comparably with the strongest hybrid model of CSwin Transformer.

- The authors further evaluate on three dense prediction tasks: ADE20K, Pascal VOC, and MS COCO. The gains of large kernels become more obvious on (i) dense tasks; and (ii) high-resolution images. In particular, SLaK-T with larger kernels (51×51) further brings 1.2% mIoU improvement over ConvNeXt-T (RepLKNet), surpassing the performance of ConvNeXt-S. On MS-COCO, SLaK fine-tuned with Cascade Mask R-CNN also outperforms the same fine-tuned ConvNeX-T by around 1% in all AP metrics.

The weakness is perhaps that the model is not as advantageous on real hardware. It’s encouraging to see the authors reported some real hardware measurements in Section 6.2, but those are 4x compared with the dense large-kernel networks (which are supposed to be extremely slow), not the off-the-shelf-competitors such as SwinT or ConvNext.

Considering the highly unconventional architecture that this paper innovates with, I won’t hold this point as a fatal weakness against this paper. However, I do suggest the authors to openly discuss this issue and acknowledge their real hardware speed limitation w.r.t. existing competitors, so that readers would get the full picture right for selecting models. A further interesting question is whether the authors could consider co-designing their novel NNs with some customizable hardware such as FPGA.


**Summary Of The Paper:**

Recent works on modern ConvNets defend the essential roles of convolution in computer vision by designing advanced architectures and plugging large kernels. This paper pushes the extreme of kernel sizes, and proposed to build a pure ConvNet model that smoothly scales up the kernel size beyond 51×51. Both empirical results and scientific insights appear to be strong.

**Summary Of The Review:**

Overall this paper looks good and the authors are encouraged to provide more discussions for the performance on real hardware.

---

> ### Author Response · Authors · 2022-11-15
> **Response to Reviewer rrkL**
>
> Thank you for leaving us such a wonderful review. We are glad that you think our paper provides strong empirical results and scientific insights. We address your comments below.
>
> **Comment 1: The weakness is perhaps that the model is not as advantageous on real hardware. It’s encouraging to see the authors reported some real hardware measurements in Section 6.2, but those are 4x compared with the dense large-kernel networks (which are supposed to be extremely slow), not the off-the-shelf-competitors such as SwinT or ConvNext. Considering the highly unconventional architecture that this paper innovates with, I won’t hold this point as a fatal weakness against this paper. However, I do suggest the authors to openly discuss this issue and acknowledge their real hardware speed limitation w.r.t. existing competitors, so that readers would get the full picture right for selecting models.**
>
> * Thank you for your thoughtful suggestion. We definitely acknowledge and would like to openly discuss this weakness. Without any sparsity-aware accelerators, our model will be slower than it looks from FLOPs.
> | Model  | Resolution | Kernel Sizes |  throughput $\uparrow$ |
> | ------------- | :-----------: |:-----------:|:-----------: |
> | Swin-T | 224$\times$224  | - | 1305 |
> | ConvNeXt-T | 224$\times$224  | 7$\times$7 | 1779 |
> | SLaK-T | 224$\times$224  | 51$\times$51 | 709 |
> * We have added the throughput of SLaK and other off-the-shelf models such as SwinT and ConvNeXt using an A100 GPU with Pytorch 1.10.0 + cuDNN 8.2.0, in FP32 precision. The results are reported in the following table. On common GPU without any sparsity-aware accelerators, the throughput of Swin and ConvNeXt is around 1.8$\times$ and 2.5$\times$ larger than SLaK, which is however much faster than we expected given the 1.3$\times$ wider model size and $>$7$\times$ larger kernel size, thanks to the kernel decomposition trick. Without using decomposition, naively scaling the kernel size of ConvNeXt to 51$\times$51 will be 8.5$\times$ slower than the 7$\times$7 kernel.
>
> * To have a better understanding, we break down the impact of each trick on throughput and report in the following table. In general, sparse large kernels cost similar run time to dense large kernels of the same size (neither acceleration nor overhead), but the decomposed kernels yield 3x real inference speed acceleration than directly using vanilla large kernels. After increasing the model width to 1.3$\times$, our model is still 2$\times$ faster than vanilla large kernels. We have added the above throughput evaluation in Appendix H.
> | Model  | Standard 51$\times$51 | Decomposed | Sparse Groups | Sparse Groups + 1.3$\times$ width|
> | ------------- | :-----------: |:-----------:|:-----------: |:-----------: |
> | SLaK-B | 96  | 283 | 282 | 198 |
>
> **Comment 2: A further interesting question is whether the authors could consider co-designing their novel NNs with some customizable hardware such as FPGA.**
> * We are very interested in exploring our sparse large-kernel neural networks on specialized hardware like CPU and FPGA, as we see much improvement room of promise here. For example, at high unstructured sparsity levels, XNNPACK (https://github.com/google/XNNPACK) has already shown significant speedups over dense baselines on smartphone processors. Moreover, the promise of dynamic sparsity has also been demonstrated on commodity hardware with CPU (https://arxiv.org/abs/1901.09181).

---

### Official Review · Reviewer_oAn8 · 2022-10-25

**Confidence:** 5
**Correctness:** 3
**Technical Novelty And Significance:** 2
**Empirical Novelty And Significance:** 2
**Recommendation:** 5

**Clarity, Quality, Novelty And Reproducibility:**

This paper is clearly presented to show the effectiveness of the proposed method with the limitation in practice. Although the number using the largest kernel ever looks interesting, the novelty seems limited due to the aforementioned reasons. The reproducibility may not be confirmed because the details of the dynamic sparsity process are not clearly presented, so I have no guarantee whether one can reach the reported score when trained from scratch.

**Strength And Weaknesses:**

Pros)
+ Overall, the paper is easy to follow.


Cons & comments)
- The novelty is limited. The following recipes to reach the goal are not new and widely-used prior arts:
   - The kernel decomposition of a large kernel into smaller ones was proposed in Inception families, and many successors adopted the idea to realize efficiency or avoid overfit.
   - Pruning and retraining (+growing) process similar to Iterative Magnitude Pruning (IMP) has also been widely used aiming the same purpose and is turned out to be very effective for maintaining precision.
   - Increasing network width has also been a regular tweak to meet the computational demand and improve the model accuracy as well since WideResNet primitively introduced it.
- Since the accuracy number in Table 3 looks promising, the accuracy gap between it and the one in Table 2 is quite large. I wonder if this is technically due to the parameterization trick in RepVGG and RepLKNet using (31x31 kernels). Specifically, Table 2 shows the performance boosts with the largest kernel size 61x61 are clearly marginal (compared with 31x31 and 51x51, +0, -0.1 for all the cases), so it seems that the proposed method does not produce a final network leveraging a larger kernel effect.
- Latency or throughput is not reported. I understand the current hardware architecture may not support the kind of sparse matrix computation, but I believe this type of work should include a speed measure.
- As the authors mention as well, the implementation of this work was done with weight masking, so using the model in practice is cumbersome, and the latency in practice may not beat the competitors'.
- The authors claim that the lack of locality is the rationale behind the performance degradation after naive training with a large kernel size (e.g., 51x51 or 61x61). However, I respectfully disagree with the claim because there is little evidence, and a more valid reason is that the model was overfitted (in Table 1). Because the training was only done with 120 epochs which is a limited training epoch; and seemingly identical data augmentation (such as the fixed RandAug coefficients, weight decay parameters, and so on) was used for all the models, so the expanded weights could be easily overfitted under weak data augmentation training regime. I recommend the authors train it for more epochs with stronger data augmentations to justify the claim. Furthermore, further backups concerning the locality should also be provided.
- The detailed procedure of the dynamic sparsity (in Figure 2) is not provided. How did the authors initialize the weights before training? How did SNIP use for giving sparsity - at a layer level? or channel level? How to tune the hyperparameters for scheduling the procedure, such as the interval of pruning/growing and the threshold of pruning?
- Particularly, the dynamic sparsity may not be universally applicable for other large-kernel networks like this method.
- How did the authors pad the feature map with such a large kernel (>56)? Zero-padded? If so, zero-padding may harm the performance due to an imbalance scale at the borders, so how did the authors handle it?
- How did the authors use the pretrained models for finetuning? All the mask was fixed, and the corresponding weights also remained frozen?



**Summary Of The Paper:**

This paper proposes a large kernel ConvNets which scaled up the kernel size beyond 51x51 up to 61x61 upon the RepVGG-like ConvNet baseline. The authors progress the idea from ConvNeXt and show the difference between the recently published RepLKNet (CVPR 2022) with the proposed model by introducing the concept of dynamic sparsity of weights consisting of sparse initialization and sparse weight training. This goes with the weight pruning and growing (i.e., adding new ones) processes during training to reach a sparse final network. This seems to be a key idea for expanding the kernel size beyond 31x31; another is decomposing a square kernel into a rectangular one (i.e., from a single NxN kernel to two successive kernels, NxM and MxN). Finally, utilizing the model with a reduced FLOP can give an opportunity of extending the network width, so it becomes a final ingredient of the proposed kernel expansion method. The authors provide ImageNet-1k evaluation compared with ConvNeXt, ViT-based models including Swin, Cswin, and DeiT. Finetuning evaluations on ADE20k, Pascal VOC2007, and MS COCO are equipped to support the effectiveness of the proposed model. Some analyses are also provided.

**Summary Of The Review:**

This paper shows the potential of using larger kernel sizes beyond 31x31 in the previous CVPR work. I'm concerned that the overall idea seems to be well organized with sub-ideas, but they are existing arts, so the novelty of this paper is limited. More specifically, one may imagine that combining a weight decomposition, training with an iterative weight pruning and growing process, and expanding the network width will naturally lead to the performance of a given network. Furthermore, the proposed method does not seem to contribute to improving the resultant model compared with the 31x31 kernel one (Table 2); but the final accuracy was reached rather leveraging the impact of the parameterization technique (Table 3). Finally, using the resultant model in practice with the trained mask is currently not straightforward for practitioners. I believe the work should need newer items and report either raised accuracies of 51x51 or 61x61 kernel models to support the main claim, so I am currently voting to reject this paper.

---

> ### Author Response · Authors · 2022-11-15
> **Response to Reviewer oAn8 (4/4)**
>
> **Comment 8: Particularly, the dynamic sparsity may not be universally applicable for other large-kernel networks like this method.**
> * Sorry but we really cannot agree on this accusation which is not based on evidence support. Firstly, we have already shown that dynamic sparsity is applicable to two prior arts large-kernel networks: ConvNeXt and RepLKNet - which are the most representative ConvNet backbones besides SLaK. Secondly, dynamic sparsity itself demonstrated effectiveness across almost every popular architecture including MLP [1], ResNet [2,3], VGG [3], ViT [4], BERT [5], RNN [6]. Therefore, we believe the dynamic sparsity is INDEED universally applicable, and we would welcome if the reviewer wants to specify this accusation more, e.g., which architecture or "other large-kernel networks" the reviewer sees as "may not be applicable" - we appreciate your question and we will be happy to clarify more if a more concrete concern can be made.
>
> **Comment 9: How did the authors pad the feature map with such a large kernel ($>$56)? Zero-padded? If so, zero-padding may harm the performance due to an imbalance scale at the borders, so how did the authors handle it?**
> * Yes, we follow the implementation of RepLKNet and simply add zero padding around input images to keep the output size consistent for the large-kernel depth-wise convolutions. Different from RepLKNet, we decompose a 51$\times$51 kernel into two 51$\times$5 and 5$\times$51 small non-square kernels which help to largely reduce the number of padding.
>
> **Comment 10: How did the authors use the pretrained models for finetuning? All the mask was fixed, and the corresponding weights also remained frozen?**
> * We clarify that we fix the masks during finetuning, so that the zero weights are kept as zero without introducing any extra parameters.
>
> [1] [Mocanu, Decebal Constantin, et al. "Scalable training of artificial neural networks with adaptive sparse connectivity inspired by network science." Nature communications 9.1 (2018): 1-12.](https://www.nature.com/articles/s41467-018-04316-3)
>
> [2] [Evci, Utku, et al. "Rigging the lottery: Making all tickets winners." ICML 2020.](https://arxiv.org/abs/1911.11134)
>
> [3] [Liu, Shiwei, et al. "Sparse training via boosting pruning plasticity with neuroregeneration." NeurIPS 2021.](https://arxiv.org/abs/2106.10404)
>
> [4] [Chen, Tianlong, et al. "Chasing sparsity in vision transformers: An end-to-end exploration." NeurIPS 2021.](https://arxiv.org/abs/2106.04533)
>
> [5] [Dietrich, Anastasia, et al. "Towards structured dynamic sparse pre-training of bert." arXiv preprint arXiv:2108.06277 (2021).](https://arxiv.org/abs/2108.06277)
>
> [6] [Liu, S., Mocanu, D. C., Pei, Y., Pechenizkiy, M. (2021, July). Selfish sparse rnn training. ICML 2021.](https://arxiv.org/abs/2101.09048)
>
> [7] [He, Kaiming, et al. "Delving deep into rectifiers: Surpassing human-level performance on imagenet classification." Proceedings of the IEEE international conference on computer vision. 2015.](https://arxiv.org/abs/1502.01852)

---

> ### Author Response · Authors · 2022-11-15
> **Response to Reviewer oAn8 (3/4)**
>
> **Comment 5: Latency or throughput is not reported. I understand the current hardware architecture may not support the kind of sparse matrix computation, but I believe this type of work should include a speed measure.\
> Comment 6: As the authors mention as well, the implementation of this work was done with weight masking, so using the model in practice is cumbersome, and the latency in practice may not beat the competitors.**
>
> * Indeed, currently the promising benefits of dynamic sparsity in our paper can not be fully translated into real speedup, as we have already thoroughly discussed in the Limitation section. Without any sparsity-aware accelerators, our model will be slower than it looks from FLOPs. Encouragingly, **Reviewer rrkL**, who also raised hardware-speedup questions, was very nice to acknowledge that for a highly unconventional architecture like ours, limited hardware support at present shouldn't be held as a fatal weakness against such.
>
> * Following your great suggestions, we have added the throughput of SLaK and other off-the-shelf models such as SwinT and ConvNeXt using an A100 GPU with Pytorch 1.10.0 + cuDNN 8.2.0, in FP32 precision without using sparsity-aware accelerators. The results are reported in the following table. On common GPU without any sparsity-aware accelerators, the throughput of Swin and ConvNeXt is around 1.8$\times$ and 2.5$\times$ larger than SLaK, which is however much faster than we expected given the 1.3$\times$ wider model size and $>$7$\times$ larger kernel size, thanks to the kernel decomposition trick. Without using decomposition, naively scaling the kernel size of ConvNeXt to 51$\times$51 will be 8.5$\times$ slower than the 7$\times$7 kernel.
> | Model  | Resolution | Kernel Size |  Throughput |
> | ------------- | :-----------: |:-----------:|:-----------: |
> | Swin-T | 224$\times$224  | - | 1305 |
> | ConvNeXt-T |  224$\times$224  | 7$\times$7 | 1779 |
> | SLaK-T |  224$\times$224  | 51$\times$51 | 709 |
> * To better understand, we break down the impact of each trick on throughput and report in the following table. In general, sparse large kernels cost similar run time to dense large kernels of the same size (neither acceleration nor overhead), but the decomposed kernels yield 3$\times$ real inference speed acceleration than directly using vanilla large kernels. After increasing the model width to 1.3$\times$, our model is still 2$\times$ faster than vanilla large kernels. We have added this discussion in Appendix H.
> | Model  | Standard 51$\times$51 | Decomposed | Sparse Groups | Sparse Groups + 1.3$\times$ width|
> | ------------- | :-----------: |:-----------:|:-----------: |:-----------: |
> | SLaK-B | 96  | 283 | 282 | 198 |
>
> * Although not the focus of our work, it would be interesting for future work to examine the speedup results of sparse large kernels, using such specialized hardware accelerators, as we see much improvement room of promise here. For example, at high unstructured sparsity levels, XNNPACK(https://github.com/google/XNNPACK) has already shown significant speedups over dense baselines on smartphone processors.
>
> **Comment 7: The detailed procedure of the dynamic sparsity (in Figure 2) is not provided. How did the authors initialize the weights before training? How did SNIP use for giving sparsity - at a layer level? or channel level? How to tune the hyperparameters for scheduling the procedure, such as the interval of pruning/growing and the threshold of pruning?**
>
> * Thanks for your comment. We actually have described the designing choices and hyperparameter tuning (such as the interval of pruning/growing and the threshold of pruning) of dynamic sparsity in Appendix G of the original submission due to the page limit. And other details about SNIP can also be found in the main paper here and there. However, we also agree that it would be better to summarize them together and be as detailed as possible. Thus, we have replaced Appendix G with full details of dynamic sparsity.
> * We also provide answers to your questions here for convenience. Regarding the initialization before training, we first initialize the dense network with the standard initialization i.e., Kaiming Uniform [7]. And for each weight, we calculate their SNIP scores as $|g \odot w|$, where $w$ and $g$ is the network weight and gradient, respectively. We globally sort all the scores across layers and force the weights with the smallest scores to be zero. Regarding the pruning and growing intervals, we conduct a lightweight sweep and choose an interval of 100 for all SLaK models as reported in Appendix G.

---

> ### Author Response · Authors · 2022-11-15
> **Response to Reviewer oAn8 (2/4)**
>
> **Comment 2: Since the accuracy number in Table 3 looks promising, the accuracy gap between it and the one in Table 2 is quite large. I wonder if this is technically due to the parameterization trick in RepVGG and RepLKNet using (31x31 kernels)...\
> Comment 3: The authors claim that the lack of locality is the rationale behind the performance degradation after naive training with a large kernel size (e.g., 51x51 or 61x61). However, I respectfully disagree with the claim because there is little evidence, and a more valid reason is that the model was overfitted (in Table 1)... I recommend the authors train it for more epochs with stronger data augmentations to justify the claim.**
>
> * We address these two comments together since they share a large overlap. We first clarify the accuracy gap between Table 2 and Table 3 due to their different training epochs (120 vs 300). We use 120-epoch training just to sketch the scaling trends of large kernel sizes due to our limited computation resources. To justify if the 120-epoch training recipe causes any overfitting to the naive kernel scaling (e.g., RepLKNet) as the reviewer recommended, we train ConvNeXt (RepLKNet) for 300 epochs with stronger data augmentation and report the results below.
> | 51$\times$51  | ConvNeXt (RepLKNet) | SLaK |
> | ------------- | :-----------: |:-----------:|
> | 100 epochs | 81.3  | 81.6 (+0.3) |
> | 300 epochs | 82.1  | 82.5 (+0.4) |
>
> * In the 300-epoch training regime, we exactly see the same performance gap between SLaK and RepLKNet. Therefore, we can conclude that (1) the 120-epoch training recipe does not cause any overfitting; (2)  our promising results in Table 3 are not due to the parameterization trick which consistently reaches inferior performance than SLaK, but are achieved by our proposed recipe.
>
> * Moreover, it is reasonable that the benefits of SLaK reach a saturation point at 61$\times$61 kernels, given that 61$\times$61 kernels are already larger than the input resolution after the 4$\times$ downsampling caused by the stem cell for 224$\times$224 ImageNet.
>
> **Comment 4: More specifically, one may imagine that combining a weight decomposition, training with an iterative weight pruning and growing process, and expanding the network width will naturally lead to the performance of a given network.**
>
> * We are a bit surprised by the "one may imagine... will naturally lead to" argument without much evidence ground, and we humbly disagree that since it seems to significantly underestimate our task's difficulty and our contributed novelty. Sharply contrary to your intuition, directly "combining" those tricks to a given network will NOT improve performance. For example, As we show in Table 2 (we also add the results here), applying each part of our proposed recipe to the original ConvNeXt with 7$\times$7 kernels leads to either marginal gains or even negative gains compared to our 51$\times$51 kernels. Therefore, the unconventionally larger kernel is the prerequisite that can unleash the power of those training techniques (a major fact that we discovered for the first time). Besides this point, we have summarized more uniqueness and novelty of our receipt that goes way beyond "naturally combining". We humbly hope the reviewer could kindly check our previous answer as other reviewers' comments, and be clarified.
> | Kernel Size  | Dense | Decomposed |  Sparse Groups | Sparse Groups + 1.3x width |
> | ------------- | :-----------: |:-----------:|:-----------: |:-----------:|
> | 7$\times$7 | 81.0  | 81.0 (+0.0) | 80.0 (-1.0) | 81.1 (+0.1) |
> | 51$\times$51 | 80.4  | 81.5 (+1.1) | 80.5 (+0.1) | 81.6 (+1.2)|

---

> ### Author Response · Authors · 2022-11-15
> **Response to Reviewer oAn8 (1/4)**
>
> We would like to thank the reviewer for the thoughtful comments and detailed comments. We provide point-wise responses to address your comments below.
>
> **Comment 1: The novelty is limited. The following recipes to reach the goal are not new and widely-used prior arts: (1) The kernel decomposition of a large kernel into smaller ones was proposed in Inception families, and many successors adopted the idea to realize efficiency or avoid overfit. (2) Pruning and retraining (+growing) process similar to Iterative Magnitude Pruning (IMP) has also been widely used aiming the same purpose and is turned out to be very effective for maintaining precision. (3) Increasing network width has also been a regular tweak to meet the computational demand and improve the model accuracy as well since WideResNet primitively introduced it.**
>
> * We politely disagree that our novelty is limited, and we respectfully suggest that the reviewer might have misunderstood our fundamental focus and novelty. We believe that our ideas, recipes, and our contributions are new and valuable for the community given our unconventional and ambitious research goal, which have been highly regarded by other reviewers: **Reviewer rrkL:** "Both empirical results and scientific insights appear to be strong", "a highly unusual architecture", "scientifically meaningful", "Novelty is overall good bold"; **Reviewer X26i**: "a novel convolutional network", "the first work that scales kernel size to 51x51", "The idea is very novel", "strong novelty"; **Reviewer YfwT**: "To the best of my knowledge the proposed approach is new", etc.
>
> * Please allow us to re-iterate the overarching goal of our paper is to understand how much potential improvement space ConvNets remain to have by purely enlarging kernel size, in the era of Transformers. Following the recent endeavors of ConvNeXt and RepLKNet that draw explosive interest from the community, we push along the direction of pure ConvNets with extremely large kernels up to 51x51 or even 61x61, which has never been discussed before. The fundamental contribution of our paper is to validate the potential of continually scaling up convolutional kernels, which is highly non-trivial as seen from our observations (e.g., the vanilla scaling of kernels beyond 31x31 using existing techniques will start to hurt the performance). Hence, each individual component of our recipe is contributing to the novel goal and unified recipe for scaling kernel size up to an unprecedented level: 51x51.
>
> * Furthermore, the usage of these components in our paper is also very different from the regular ones as you mentioned. (1) While kernel decomposition was introduced in Inception families, the possibility of using kernel decomposition to scale up kernel size up to 51x51 or 61x61 has never been explored. We for the first time unveil that using kernel decomposition can surprisingly reverse the accuracy descending trend of scaling kernels beyond 31x31, i.e., the decomposed 51x51 kernels achieve better performance than 31x31. (2) Dynamic sparsity is proposed and widely used for model compression, i.e., finding **smaller subnetworks** that perform close to the dense networks. We for the first time explore the possibility of dynamic sparsity as a neural network scaling principle to build **larger networks**, suggesting that sparsity, as the "old friend" of model compression, can make a promising tool to boost the complementary goal of scaling up.

---

> ### Author Response · Authors · 2022-11-22
> **We are keen to discuss further with you**
>
> Dear Reviewer **oAn8**,
>
> We thank you for your time and constructive comments. We really hope to discuss further with you to see if our response solves your concerns.
>
> In our response, we have highlighted our research goal and novelty of our paper; provided detailed responses and experiments to your comments.
>
> We genuinely hope reviewer **oAn8** could kindly check our response. Thank you!
>
> Best wishes,
>
> Authors

---

> ### Author Response · Authors · 2022-11-26
> **We are keen to discuss further with you**
>
> Dear Reviewer **oAn8**,
>
> We thank you for your time and your constructive comments.
>
> As the discussion period is approaching its end, we would really appreciate it if you could kindly let us know whether there are any further questions. We will be more than happy to address them.
>
> Best wishes,
>
> Authors

---

> > ### Author Response · Authors · 2022-12-07
> > **One last reminder for discussion**
> >
> > Dear Reviewer oAn8,
> >
> > As we still did not hear back from you, we continue to stand by for any updated opinion you might have, **after our point-to-point rebuttal, as well as other reviewers' responses which have now turned unanimously positive/supportive.**
> >
> > We hope you will get a moment to check those and let us know if there is any further concern.
> >
> > Authors.

---

### Public Comment · ~Timothée_Masquelier1 · 2022-11-06
**About number of FLOPs and parameters**

Hello,

I'm not a reviewer for this paper, but I read it in detail as soon as it came out on arXiv, and I've found the approach extremely appealing :-)

However, I have two important concerns:

1) I suspect the FLOPs reported here do not include the ones of the gemm method. When including them, the total number of FLOPs could be much higher than the ConvNeXt baselines to which they compare themselves.

2) The reported numbers of parameters only include the non-zero ones (https://github.com/VITA-Group/SLaK/issues/12). I think the total number of parameters should also be reported. It is the total number of parameters that matters for 99% of the users, who will run the code on GPUs which do not leverage sparsity.

Best,

---

> ### Author Response · Authors · 2022-11-06
> **Thanks and response to concerns**
>
> Hi Timothée Masquelier,
>
> Thanks for your enthusiasm for our paper and sharing your concerns with us. Indeed, sparse large kernels are not fully supported in current common GPUs without sparsity-aware accelerators. Due to this reason, we choose to implement our models with binary masks, as we have mentioned in the Limitation section.
>
> However, we would like to clarify that the FLOPs and parameter count reported in our paper are correctly measured by counting the total number of multiplications and additions layer by layer (including the large-kernel convolutional layers implemented by GEMM) for a given layer sparsity, following the previous art of model pruning and sparse training (Rethinking Pruning https://arxiv.org/pdf/1810.05270.pdf, RigL https://arxiv.org/pdf/1911.11134.pdf, and Top-KAST https://arxiv.org/abs/2106.03517). Similarly, the parameter count is measured as the number of parameters in use - non-zero parameters. Given that we use one set of hyperparameters (width 1.3x and sparsity 0.4) for all models, it is relatively easy to calculate the total number of parameters. We will include the total number of parameters (including the zeroed and non-zeroed) in our revision.
>
> Although not the focus of our current work, it would be interesting for future work to examine the speedup results of sparse large kernels, using such specialized hardware accelerators, as we see much improvement room of promise here. For example, at high unstructured sparsity levels, XNNPACK (https://github.com/google/XNNPACK) has already shown significant speedups over dense baselines on smartphone processors.

---

> > ### Public Comment · ~Timothée_Masquelier1 · 2022-11-08
> > **Thanks and one more question for the FLOPs**
> >
> > Thanks a lot for your answer!
> >
> > OK for the parameters.
> >
> > For the FLOPs, can I ask you what code has been used to measure FLOPs? Is it via this library (flops-counter) https://github.com/sovrasov/flops-counter.pytorch ?

---

> > > ### Author Response · Authors · 2022-11-08
> > > **Regarding FLOPs**
> > >
> > > Sure. We did not use the flops-counter library as you mentioned.
> > >
> > > For the FLOPs, we adopted the THOP: [PyTorch-OpCounter](https://github.com/Lyken17/pytorch-OpCounter) and adjusted the FLOPs measurement for Conv and linear layers based on their layer sparsity, following previous art [rethinking-network-pruning, ICLR2019](https://github.com/Eric-mingjie/rethinking-network-pruning/blob/master/cifar/weight-level/count_flops.py). To eliminate the possible missing of FLOPs caused by the DepthWiseConv2dImplicitGEMM, we replace the GEMM Conv layers with the standard torch.nn.Conv2d layers when measuring FLOPs. Thus, we are confident that our measurement of FLOPs is 100% correct. Thanks.

---

> > > > ### Public Comment · ~Timothée_Masquelier1 · 2022-11-09
> > > > **All clear**
> > > >
> > > > Oh, I see, your FLOPs also take sparsity into account!
> > > > All clear now.
> > > > Thanks a lot for your answers!

---

### Decision · Program_Chairs · 2023-01-20

**Decision:**

Accept: poster

**Justification For Why Not Higher Score:**

Unclear if gains will be realized on accelerated hardware

**Justification For Why Not Lower Score:**

Simple method, strong empirical results.

**Metareview: Summary, Strengths And Weaknesses:**

In this work, the authors propose a method for increasing the size of the receptive fields of convolutional kernels in CNNs without incurring the extra penalty in terms of computational performance. In particular, the authors achieve this result by employing dynamic sparsity on non-square filters. The authors show favorable ImageNet classification performance, semantic segmentation on ADE20K, object detection on Pascal VOC and object detection/segmentation on COCO. These positive results indicate that employing large, sparse receptive fields for CNN convolutional kernels may be applied across a wide array of computer vision problems.

The reviewers commented on the clarity of presentation, the thorough and strong experimental results and the simplicity of the proposed method. The reviewers found it particularly interesting that it is possible to train on large kernels (e.g. 51x51) without modifications, but is especially well leveraged by enforcing a factorized, non-square sparsity across the kernel. The primary concern expressed by reviewers was the central focus and emphasis of the paper. For instance, one reviewer points out that the central idea of the paper is the use of filters with non-square filters at large sizes (e.g. cross-shaped filters, etc.). Subsequent discussion arrived at an improved description of the contribution of the method. Given the strong results, the simplicity of the method and the implications for the larger computer vision community, this paper will be accepted into this conference.


**Note From Pc:**

if the above contains the word "oral" or "spotlight" please see: "oral" presentation means -> notable-top-5% and "spotlight" means -> notable-top-25%. As stated in our emails, we are disassociating presentation type from AC recommendations